# Generalizing Single-Frame Supervision to Event-Level Understanding for Video Anomaly Detection

**Junxi Chen[1]**   **Liang Li[2]**[*]   **Yunbin Tu[1]**   **Li Su[1]**[*]   **Zhe Xue[3]**   **Qingming Huang[1,2]**

[1]University of Chinese Academy of Sciences
[2]Key Laboratory of Intelligent Information Processing, ICT, CAS
[3]Beijing University of Posts and Telecommunications
`{chenjunxi22, tuyunbin22}@mails.ucas.ac.cn`   `liang.li@ict.ac.cn`
`{suli, qmhuang}@ucas.ac.cn`   `xuezhe@bupt.edu.cn`

## Abstract

Video Anomaly Detection (VAD) aims to identify abnormal frames from discrete events within video sequences. Existing VAD methods suffer from heavy annotation burdens in fully-supervised paradigm, insensitivity to subtle anomalies in semi-supervised paradigm, and vulnerability to noise in weakly-supervised paradigm. To address these limitations, we propose a novel paradigm: Single-Frame supervised VAD (SF-VAD), which uses a single annotated abnormal frame per abnormal video. SF-VAD ensures annotation efficiency while offering precise anomaly reference, facilitating robust anomaly modeling, and enhancing the detection of subtle anomalies in complex visual contexts. To validate its effectiveness, we construct three SF-VAD benchmarks by manually re-annotating the ShanghaiTech, UCF-Crime, and XD-Violence datasets in a practical procedure. Further, we devise Frame-guided Progressive Learning (FPL), to generalize sparse frame supervision to event-level anomaly understanding. FPL first leverages evidential learning to estimate anomaly relevance guided by annotated frames. Then it extends anomaly supervision by mining discrete abnormal events based on anomaly relevance and feature similarity. Meanwhile, FPL decouples normal patterns by isolating distinct normal frames outside abnormal events, reducing false alarms. Extensive experiments show SF-VAD achieves state-of-the-art detection results while offering a favorable trade-off between performance and annotation cost. The benchmarks and code are available at `https://github.com/Junxi-Chen/SF-VAD`.

## 1 Introduction

Video Anomaly Detection (VAD) aims to identify abnormal frames within video sequences, which attracts substantial research attention due to its broad applicability in critical areas, e.g., public security [34], traffic surveillance [23], malicious content moderation [11]. However, precisely identifying anomalies remains challenging. First, anomalies inherently span across multiple temporally disjoint events, hindering the learning of a unified and robust anomaly pattern. Second, abnormal events may exhibit low salience, blending with context, thus complicating their discrimination from visual noise.

To tackle the challenges, existing methods employ three primary VAD paradigms, as illustrated in Figure 1a: fully-supervised, semi-supervised, and weakly-supervised VAD. ***Fully-supervised*** VAD (Fully-VAD) approaches [20, 16] require dense frame-level annotations. However, the inherent ambiguity of anomaly boundaries leads to inconsistent labeling, which impairs the learning of coherent decision boundaries. The burden of extensive annotations also limits its generalizability. ***Semi-supervised VAD*** (Semi-VAD) methods [26, 31, 61, 60] assume access only to normalcy, and

---

[*]Corresponding authors.

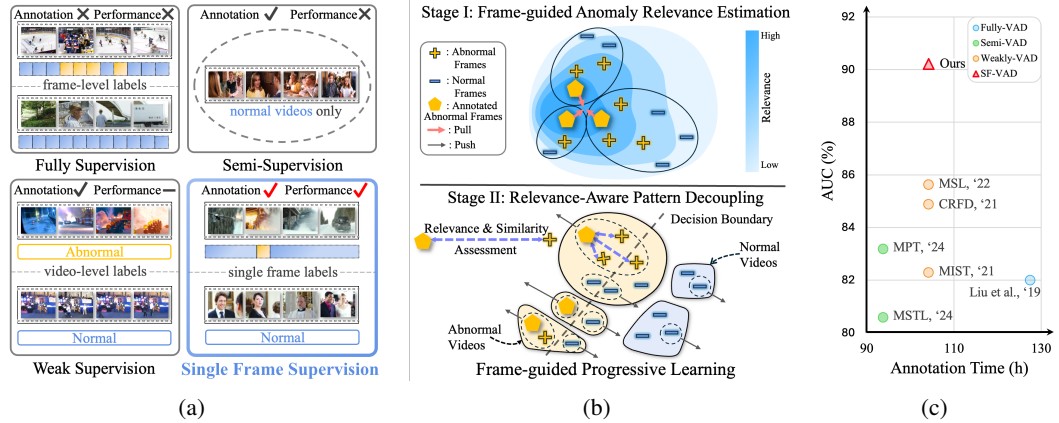

Figure 1: (a) Video Anomaly Detection (VAD) paradigms. The proposed Single-Frame supervised VAD (SF-VAD) paradigm offers fine-grained anomaly guidance by a single abnormal frame annotation, while ensuring annotation efficiency. (b) Frame-guided Progressive Learning (FPL). FPL initially estimates the anomaly relevance to the annotated frames in Stage I. It then generalizes anomaly supervision from one frame to intact intervals and disentangles normal context by exploiting anomaly relevance and feature similarity in Stage II. (c) Performance versus annotation time on UCF-Crime. SF-VAD demonstrates an ideal trade-off between performance and annotation efficiency.

identify anomalies as deviation from learned normal patterns. However, it may fail to capture trivial anomalies that exhibit low visual salience and closely resemble normal behaviors, resulting in limited robustness. Alternatively, *Weakly-supervised VAD* (Weakly-VAD) [57, 5, 47, 58] utilizes video-level annotations to model distinctive cues to anomalies by separating prominent features of normal and abnormal videos, which delivers a superior detection precision. Even so, due to disruptive normal context, and lack of precise anomaly reference, such paradigm struggles to model noise-free anomaly patterns and suffers from severe false alarms, leading to a bottleneck for advances in VAD.

To expand the frontier of VAD, we propose an annotation-efficient paradigm: ***Single-Frame supervised VAD*** (SF-VAD), which is annotation-efficient, leveraging abnormal videos with *single* abnormal frame labels, as illustrated in Figure 1a. Correspondingly, to evaluate its effectiveness in real-world scenarios, we construct three SF-VAD benchmarks by manually re-annotating existing VAD datasets [21, 34, 46] in practical procedure. For annotation efficiency, annotators are allowed to freely navigate within videos to label a random abnormal frame per abnormal video, thereby alleviating the need for full video review and exhaustive temporal localization. Moreover, single-frame supervision offers fine-grained anomaly guidance. On the one hand, SF-VAD provides explicit references to various abnormal behaviors, not only facilitating noise-robust anomaly modeling but also accentuating subtle anomalies from its context, e.g., shoplifting and abuse, which are temporally brief and visually inconspicuous. On the other hand, frame supervision can act as a contrastive anchor to distinguish normal context, reducing false alarms by suppressing pseudo anomalies that contradict annotated behaviours. Nevertheless, the effective application of single-frame supervision to VAD involves two main challenges: (1) bridging the gap between sparse single frames and temporally extended abnormal events across multiple intervals; (2) disentangling normal contextual patterns from truly anomalous behaviors within the anomaly domain.

To this end, we propose a novel framework named Frame-guided Progressive Learning (FPL), which (1) progressively generalizes sparse single-frame anomaly supervision to discrete temporal intervals of abnormal events; and (2) effectively disentangles distinct normal patterns from abnormal video segments, as illustrated in Figure 1b. Concretely, FPL consists of two stages: In stage I, we devise frame-guided anomaly relevance estimation, which leverages the theory of evidence [14, 32] to quantify the relevance of each frame to the annotated abnormal frames. This relevance estimation serves as a foundation for bridging the gap between sparse frame-level supervision and holistic event-level anomaly localization. In stage II, FPL performs relevance-aware pattern decoupling, aiming to disentangle heterogeneous patterns within and outside abnormal events. For anomaly modeling, we tailor an anomaly event mining algorithm that identifies temporally discrete abnormal intervals by jointly considering anomaly relevance scores and frame-wise feature similarities. To decouple

normal patterns embedded in abnormal videos, FPL isolates normalcy from two key sources: (1) the pre-event interval, which typically manifests undisturbed contextual normal cues before the onset of anomalies; and (2) frames with the lowest anomaly probability in the post-event interval, which often reflect unique normal characteristics within the anomaly domain, signaling a re-stabilization state following the abnormal episode. Through this two-stage process, FPL fully leverages sparse single-frame supervision to enable comprehensive event-level anomaly understanding, while reducing false alarms via disentanglement of normal frames beyond anomalous intervals. Extensive experiments manifest that the proposed SF-VAD paradigm reveals a favorable trade-off between performance and annotation cost, as depicted in Figure 1c.

Our main contributions are as follows:

- We propose a novel and annotation-efficient single-Frame supervised VAD paradigm, which provides fine-grained guidance for modeling anomalous behavior. To facilitate future research, we construct and release three SF-VAD datasets.

- We devise a frame-guided progressive learning framework, which first estimates the anomaly relevance, then progressively generalizes anomaly supervision from single frames to multiple abnormal events, and meanwhile decouples distinct normal patterns.

- Extensive results show our SF-VAD method achieves state-of-the-art performance, including a higher AUC and a lower false alarm rate.

## 2   Related Work

**Weakly-supervised Video Anomaly Detection**    Given the close relation between SF-VAD and Weakly-VAD, we review mainstream methods under the weak supervision setting. Most Weakly-VAD methods employ MIL, whose objective can be denoted as:

$$\mathcal{L}_{\text{MIL}} = -\log(\frac{1}{k}\sum_{i\in \text{top-k}} \hat{y}_i^+) - \log\left(1 - \max\left(\hat{\boldsymbol{y}}^-\right)\right), \tag{1}$$

where $\hat{y}_i^+$ indicates the anomaly score of the $i$-th frame in abnormal videos and $\hat{\boldsymbol{y}}^-$ refers to the anomaly scores of normal videos. However, the reliance on top-k sampling in MIL often introduces noisy samples, which can destabilize training and couple opposite patterns. To alleviate the problem, some works [6, 64, 37] explore feature decoupling mechanisms. Tian et al. [37] improve the feature disparity by feature magnitude, and sample hard normal instances to highlight abnormal features contrastively. Chen et al. [6] further propose a scene-adaptive magnitude contrastive mechanism that promotes intra-class feature similarity while enlarging inter-class separation. While other works [18, 57] explore a more stable training process to alleviate the impact of noise. Li et al. [18] propose a two-stage self-training framework to model the temporal continuity of abnormal events by sequence-aware pseudo labels. Later, Zhang et al. [57] further exploit the event completeness and prediction uncertainty in self-training framework to capture intact abnormal patterns, while alleviating the impact of noise. Motivated by the success of multi-modal approaches [17, 55, 39, 8, 38], some recent works [5, 49, 47] also explore video anomaly detection leveraging multi-modal cues.

**Inexact Supervision in Computer Vision**    Single-frame supervision falls under the category of inexact supervision, which delivers an ideal performance-cost trade-off in a variety of computer vision tasks [51, 62, 56] by learning from imprecisely labeled data. In semantic segmentation, Bearman et al. [2] first introduce point supervision, which incorporates a generic objectness prior to reveal objects from the background and delivers promising performance. Later, Mettes et al. [25] extend the idea to frame supervision in video spatio-temporal action localization. Ma et al. [24] further introduce frame supervision to temporal action localization, and a corresponding SF-Net, which models actionness by annotated frames and a background mining algorithm to decouple irrelevant frames. Li et al. [19] propose a frame-supervised temporal action segmentation method that leverages all frames with a confidence loss based on the distance to the annotated frame. Cui et al. [9] introduce frame-level supervision into language-driven moment retrieval by modeling the probability distribution of foreground action behaviors using a Gaussian distribution centered on the annotated frame. Recently, Zhang et al. [59] introduced multi-frame supervision to the VAD task, demonstrating the effectiveness of frame-level supervision. However, it suffers from a heavy annotation burden due to the need for full video review. Detailed discussion are provided in the Appendix K.

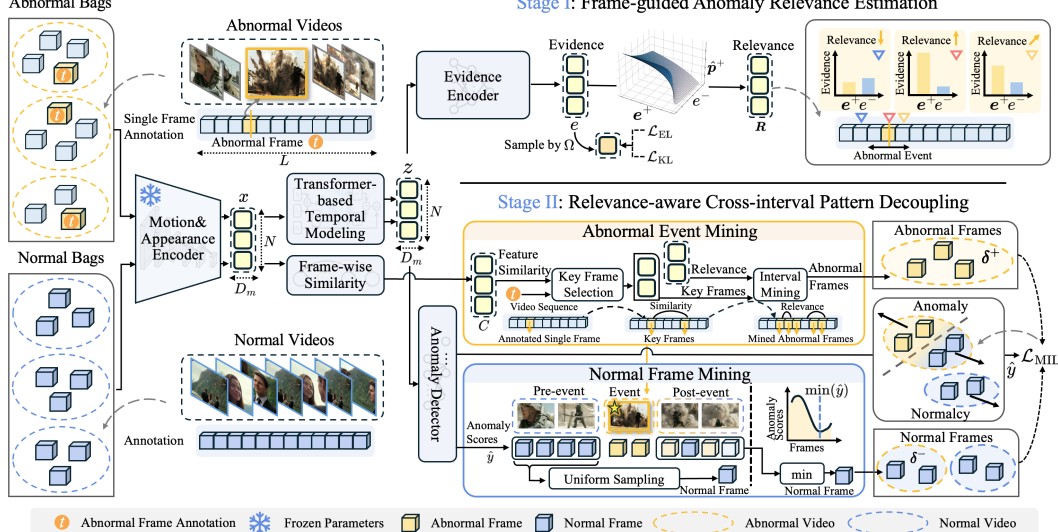

Figure 2: Overview of Frame-guided Progressive Learning framework. First, unified representations are extracted by frozen motion and appearance encoder, followed by a transformer based temporal modeling module. In stage I, Frame-guided Anomaly Relevance Estimation is performed by evidential learning to ensure a reliable anomaly probing. In Stage II, we conduct Relevance-aware Cross-interval Pattern Decoupling, where abnormal events are mined and learned based on anomaly relevance and feature similarity to the annotated frame, while opposite normal patterns are subsequently separated to disentangle the abnormal and normal patterns.

## 3 Method

### 3.1 Problem Formulation

Single-Frame supervised VAD (SF-VAD) leverages videos with single-frame annotations to learn anomalies and predicts frame-level anomaly scores to identify abnormal frames. Specifically, during training, an abnormal sample is defined as $\{\boldsymbol{v}^+, t\}$, where $\boldsymbol{v}^+$ denotes an abnormal video containing at least one abnormal frame, and $t \in \mathbb{R}$ refers to the index of the annotated abnormal frame. Correspondingly, a normal sample is $\{\boldsymbol{v}^-, 0\}$, where label $0$ indicates video $\boldsymbol{v}^-$ containing only normal frames. During inference, given a video $\boldsymbol{v}$ with $L$ frames, the goal is to predict fine-grained anomaly scores $\hat{\boldsymbol{y}} \in \mathbb{R}^L$ to align with the ground-truth $\boldsymbol{y} \in \{0,1\}^L$, where $y_i = 1$ indicates that the $i$-th frame is abnormal and $y_i = 0$ indicates that the $i$-th frame is normal.

### 3.2 Method Overview

As depicted in Figure 2, the proposed frame-guided progressive learning framework comprises two stages that gradually generalize single frame guidance for abnormal event modeling and normalcy decoupling in more complete temporal scope. As in Section 3.3, stage I, termed frame-guided anomaly relevance estimation, aims to establish reliable reference to guide the subsequent temporal abnormal event analysis. Specifically, it employs evidential learning, grounded in the theory of evidence [14, 32], to estimate both anomaly probability and uncertainty of prediction. In stage I, the anomaly learning is exclusively optimized by annotated abnormal frames, thus the predicted frame-wise uncertainty reflects the degree of deviation from the learned noise-free abnormal patterns, thereby quantifying the anomaly relevance. Subsequently, as in Section 3.4, we perform relevance-aware pattern decoupling in stage II, which extends single frame supervision across multiple disjoint abnormal events guided by estimated anomaly relevance and frame-wise similarity, while constantly decoupling normal context features outside abnormal events.

Concretely, given a video $\boldsymbol{v}$, it is split into $N$ non-overlapping clips to mitigate temporal redundancy present in video media. Then, the model encodes its motion and appearance features into a unified representation, denoted as $\boldsymbol{x} \in \mathbb{R}^{N \times D_m}$, where $D_m$ represents the feature dimension, taking

advantage of pre-trained encoders [30, 3]. Following well-established temporal modeling architectures [64, 29, 47], we employ a Transformer-based Temporal Modeling module to capture anomalies' multi-scale temporal regularities $z \in \mathbb{R}^{N \times D_m}$. Building upon the temporal features, multi-layer convolutional networks are employed as evidence encoder and anomaly detector to predict evidence for relevance estimation in Stage I and infer the final anomaly scores in Stage II.

### 3.3 Frame-guided Anomaly Relevance Estimation

In stage I, we strive to measure the anomaly relevance to annotated abnormal frames, which functions as reliable reference for subsequent abnormal event mining. We observe that annotated frames manifest the high visual consistency with their surrounding context, which undermines the discriminability of feature similarity in short temporal scope. To overcome the limitation, we propose Frame-guided Anomaly Relevance Estimation (FARE) by formulating evidential learning [32] and multi-instance learning [34]'s training scheme for SF-VAD. Specifically, the model incorporates an evidence encoder that learns abnormal evidence exclusively from annotated abnormal frames. Since this evidence is distilled directly from precise abnormal cues, it preserves the noise-free and core characteristics of the anomalous event. Consequently, the evidence can be utilized to quantify the relevance to anomalies.

Regarding VAD as a binary frame classification problem, the objective of evidential learning is to estimate the classification probability $p = [p^+, p^-]$ for a given video $v$, where $p^+, p^- \in \mathbb{R}^N$ indicate the abnormal and normal probability respectively. In this formulation, $p$ follows Bernoulli distribution, with $p^+ + p^- = 1$. Evidential learning models the Beta distribution, the conjugate prior of the Bernoulli probability $p$, to quantify the amount of evidence supporting each prediction, enabling the model to express varying degrees of confidence in its outputs. The parameters of the Beta distribution $\alpha \in \mathbb{R}^{N \times 2}$ are derived from the evidence learned by the model. A dedicated evidence encoder network, $g(v \mid \theta)$, with parameters $\theta$, is designed to predict the evidence $e \in \mathbb{R}^{N \times 2}$ that supports the inference for each class. These evidence values are directly related to the parameters of the Beta distribution by $\alpha = e + 1$. The total evidence, also known as the Beta strength, for each frame is represented by $S \in \mathbb{R}^N$, calculated as the sum of the Beta parameters:

$$S = \sum_{e \in \{e^+, e^-\}} (e + 1), \tag{2}$$

where $e^+, e^-$ denote evidence for anomaly and normality respectively. Under these circumstances, a greater value of $S$ signifies more accumulated evidence supporting the prediction, thereby indicating a lesser prediction uncertainty. The expected anomaly probability $\hat{p}^+ \in \mathbb{R}^N$ can be calculated by evidence and total evidence:

$$\hat{p}^+ = \frac{e^+ + 1}{S}. \tag{3}$$

In the absence of fine-grained frame-level labels, our SF-VAD method adopts the same multi-instance learning training scheme, which is widely adopted in weakly-VAD methods [34, 6, 64]. To guarantee a noise-free anomaly relevance estimation, the anomaly representations are learned by annotated abnormal frames exclusively in stage I. Instance $v_i$ is sampled from frame set $\Omega$, which comprises only annotated abnormal frames and the prominent normal frames:

$$\Omega = \{v_i^+, v_j^- \mid i = \phi(t), j = \operatorname{argmin}(\hat{p}^+)\}, \tag{4}$$

where $\phi$ refers to the map function from annotation to video clip index. The evidential learning target of instance $v_i$ can be denoted as:

$$\mathcal{L}_{\text{EL}}^i(\theta) = y_i(\psi(S_i) - \psi(e_i^+ + 1)) + (1 - y_i)(\psi(S_i) - \psi(e_i^- + 1)), \tag{5}$$

where $\psi$ refers to the digamma function. To prevent the overconfidence of prediction and better encourage uncertainty learning, a regularization term is further introduced. The leading evidence from predicted Beta distribution parameters is removed, yielding $\tilde{\alpha} \in \mathbb{R}^N$:

$$\tilde{\alpha} = \mathbf{y} + (1 - \mathbf{y}) \odot \alpha, \tag{6}$$

where $\mathbf{y}$ denotes the one-hot encoded ground-truth label. Then the regularization constraint encourages $B(\mathbf{p} \mid \tilde{\alpha})$ to follow the uniform Beta distribution. For instance $v_i$, the regularization constraint can be denoted as:

$$\mathcal{L}_{\text{KL}}^i(\theta) = D_{\text{KL}}\left[B\left(\mathbf{p}_i \mid \tilde{\boldsymbol{\alpha}}_i\right) \| B\left(\mathbf{p}_i \mid \mathbf{1}\right)\right]$$

$$= \log\left(\frac{\Gamma(\sum_{\tilde{\alpha}_i \in \{\tilde{\alpha}_i^+, \tilde{\alpha}_i^-\}} \tilde{\alpha}_i)}{\prod_{\tilde{\alpha}_i \in \{\tilde{\alpha}_i^+, \tilde{\alpha}_i^-\}} \Gamma(\tilde{\alpha}_i)}\right) + \sum_{\tilde{\alpha}_i \in \{\tilde{\alpha}_i^+, \tilde{\alpha}_i^-\}} (\tilde{\alpha}_i - 1)\left[\psi(\tilde{\alpha}_i) - \psi(\sum_{\tilde{\alpha}_j \in \{\tilde{\alpha}_j^+, \tilde{\alpha}_j^-\}} \tilde{\alpha}_j)\right], \tag{7}$$

where $B$ represents Beta distribution, $\Gamma(\cdot)$ is the gamma function and $D_{\text{KL}}$ indicates Kullback-Leibler divergence. The overall optimization target $\mathcal{L}_{\text{FARE}}$ in stage I can be expressed as follows:

$$\mathcal{L}_{\text{FARE}} = \sum_{i=1}^{N} \mathcal{L}_{\text{EL}}^i(\theta) + \lambda_n \sum_{i=1}^{N} \mathcal{L}_{\text{KL}}^i(\theta), \tag{8}$$

where $\lambda_n$ is the annealing coefficient corresponding to training epoch $n$. Lastly, the anomaly relevance $\boldsymbol{R} \in \mathbb{R}^N$ can be quantified as:

$$\boldsymbol{R} = 1 - \frac{2}{\boldsymbol{S}}. \tag{9}$$

This formulation establishes an aligned relationship between beta strength and relevance, where greater total evidence corresponds to higher relevance. Since Stage I exclusively models abnormal patterns based on annotated frames, a higher anomaly relevance indicates a stronger association with the annotated abnormal distribution.

## 3.4 Relevance-aware Pattern Decoupling

---
**Algorithm 1** Abnormal Event Mining

---
**Input:** Frame annotation $t$, Feature similarity $C$, Anomaly Relevance $R$, Thresholds $\theta$
**Output:** Set of abnormal frame numbers $\delta^+$
1: $\delta^+ = \{\}, \zeta = \{\phi(t)\}$
2: **if** $\text{var}(C) > \theta_1$ **then**
3:     $\zeta \leftarrow \zeta \cup \{i \mid C_i > \theta_2, |i - j| > \theta_3 N\}$
4: **end if**
5: **for** each $i \in \zeta$ **do**
6:     **if** $C_i < \Theta$, where $\Pr[C > \Theta] \leq 0.1$ **then**
7:         $l = \max\{i \mid 1 \leq i \leq t, S[i] > S[i+1]\}$
8:         $r = \min\{i \mid t \leq i \leq N, S[i] < S[i+1]\}$
9:         $\delta^+ \leftarrow \delta^+ \cup \{i \mid l \leq i \leq r\}$
10:    **else**
11:        $\delta^+ \leftarrow \delta^+ \cup \{i\}$
12:    **end if**
13: **end for**
14: **return** $\delta^+$

---

In stage II, we perform relevance-aware pattern decoupling to expand anomaly supervision from one frame to multiple anomalous events by estimated anomaly relevance and frame-wise similarity, while decoupling disruptive normal context. The frame-wise cosine similarity $\boldsymbol{C} \in \mathbb{R}^N$ with respect to the annotated abnormal clip is computed as follows:

$$\boldsymbol{C} = \left[\frac{\boldsymbol{x}_{\phi(t)} \cdot \boldsymbol{x}_i}{\left\|\boldsymbol{x}_{\phi(t)}\right\|_2 \left\|\boldsymbol{x}_i\right\|_2}\right]_{i \in [1,N]}, \tag{10}$$

where $\|\cdot\|_2$ refers to L2 Norm. On the one hand, feature similarity is limited for interval mining due to strong temporal coherence within local neighborhoods. However, high similarity across temporally distant frames often reveals recurring abnormal patterns, making it a reliable cue for key frame discovery. On the other hand, the estimated anomaly relevance captures continuous temporal cues within abnormal events. As in Figure 2, high relevance emerges within abnormal events which are strongly aligned with the learned distribution of annotated anomalies. Conversely, frames near event boundaries tend to exhibit transitional patterns that deviate from both distinct normal and abnormal distributions modeled in StageI, leading to lower relevance. By leveraging these relevance dynamics, FPL anchors on key frames and progressively expands to the full temporal extent of abnormal events.

Harnessing their complementary strengths, we devise an abnormal interval mining algorithm that first probes key frames located in multiple abnormal events via feature similarity, and then expands such key frames to abnormal event intervals inferred by anomaly relevance, as depicted in Algorithm 1. Concretely, by measurement of variance of the similarity, the similarity magnitude, and the distance to annotated frame, we gain the key abnormal frame set $\zeta$. Then we generally expand the abnormal frames in $\zeta$ to entire frames in abnormal events $\delta^+$ by concave interval of anomaly relevance.

Furthermore, FPL decouples disruptive normal context patterns outside abnormal events. An abnormal video typically encompasses three temporal phases: pre-event phase (the period preceding the anomalous event), event phase (during the occurrence of the anomaly), and post-event phase (the aftermath following the event). One the one hand, FPL uniformly samples frames from pre-event phase, which reveals distinct normal contextual cues unaffected by anomalies. On the other hand, the content in post-event phase contains unique normal characteristics within anomaly domain, reflecting a re-stabilized state following abnormal events. Yet, such phase contains visually noisy frames that may still bear traces of anomaly, e.g., the smoke and flame after "explosion", the chaos after "riot", the stasis after "car accident". Thereby, we sampled frames with the lowest anomaly scores in post-event phase, which contributes valuable negative examples to the anomaly learning process by identifying the most pronounced normal-like characteristics in potentially noisy regions. Hence, the normal frames set $\delta^-$ in abnormal video, from which we decouple normal patterns, can be denoted as:

$$\delta^- = \{\lfloor i \rceil, j \mid i \sim U(1, \min(\delta^+)), \ j = \operatorname*{argmin}_{k > \max(\delta^+)} (\hat{y}_k)\}, \tag{11}$$

where $U$ indicates uniform distribution.

A multi-layer convolutional network is applied as anomaly detector to predict anomaly scores $\hat{y} \in \mathbb{R}^N$ by temporal features $z$. Leveraging both abnormal frame set $\delta^+$ and normal frame set $\delta^-$, the final $\mathcal{L}_{\text{MIL}}$ is computed by cross-entropy:

$$\mathcal{L}_{\text{MIL}} = \sum_{i \in \delta^+} \log \hat{y}_i^+ + \sum_{i \in \delta^-} \log \left(1 - \hat{y}_i^+\right) + \lambda \log(1 - \max(\hat{\boldsymbol{y}}^-)), \tag{12}$$

where $\hat{y}^+, \hat{y}^-$ denote anomaly scores for abnormal and normal videos respectively and $\lambda$ corresponds to balancing coefficient.

## 4 Experiments

### 4.1 Dataset Construction

To validate the effectiveness of proposed paradigm, we construct three high-quality, human-annotated SF-VAD datasets based on the public benchmarks: ShanghaiTech [21], UCF-Crime [34], and XD-Violence [46]. Annotations are obtained via a crowdsourcing platform following a practical and efficient labeling protocol where annotators are asked to label the first abnormal frame they identify in each abnormal video. To simulate natural viewing behaviors, they are allowed to freely navigate the video, including skipping segments or adjusting playback speed, without being required to review full video. This protocol maximizes annotation efficiency and accounts for the inherent randomness in which abnormal events first attract attention, while avoiding bias introduced by artificial constraints. As a result, it provides a more faithful assessment of single-frame supervision in realistic settings. Details are provided in the Appendix B.

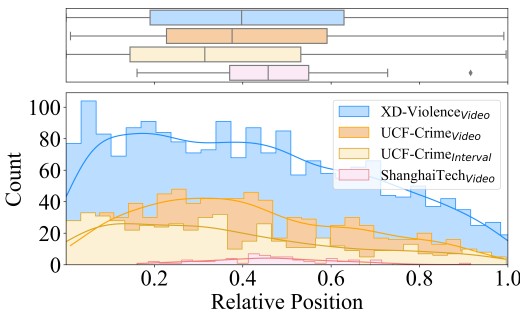

Figure 3: Visualization of dataset statistic.

### 4.2 Dataset Statistic

We analyze the temporal distribution of annotated frames in the SF-VAD benchmarks, as illustrated in Figure 3. The y-axis represents the count of annotations, while the x-axis indicates the relative

Table 1: Performance comparison with state-of-the-art methods.

| Supervision | Methods | Text | Feature | XD(%) | SH(%) | UCF(%) |
|---|---|---|---|---|---|---|
| Fully-. Supervised | ARG$_{MM\ '19}$ [20] | - | NLN | - | - | 82.0 |
| | Our Baseline | - | I3D RGB | - | - | 85.52 |
| Semi- Supervised | SVM Baseline | - | I3D+VGGish | 50.78 | - | - |
| | SCR$_{MM\ '20}$ [35] | - | - | - | 74.70 | 72.7 |
| | Conv-AE$_{CVPR\ '16}$ [12] | - | I3D+VGGish | 30.77 | - | 50.60 |
| | LANP$_{ECCV\ '24}$ [33] | - | I3D RGB | - | 88.32 | 80.02 |
| | MGEnet$_{MM\ '24}$ [48] | - | Video Swin | - | 86.9 | - |
| | AED-MAE$_{CVPR\ '24}$ [31] | - | - | - | 79.1 | - |
| | MULDE [26]$_{CVPR\ '24}$ | - | Hiera-L | - | 81.3 | 78.50 |
| Weakly- Supervised | MIL-Rank [34]$_{CVPR\ '18}$ | - | C3D RGB | 73.20 | 86.30 | 75.41 |
| | CA-VAD$_{TMM\ '21}$ [4] | - | I3D RGB | 76.90 | 92.25 | 84.62 |
| | RTFM$_{ICCV\ '21}$ [37] | - | I3D RGB | 77.81 | 97.21 | 84.30 |
| | CRFD$_{TIP\ '21}$ [44] | - | I3D RGB | 75.90 | 97.48 | 84.89 |
| | MSL$_{AAAI\ '22}$ [18] | - | VideoSwin | 78.59 | 97.32 | 85.62 |
| | S3R$_{ECCV\ '22}$ [43] | - | I3D RGB | 80.26 | 97.48 | 85.99 |
| | CMA-LA$_{ICCECE\ '22}$ [28] | - | I3D+VGGish | 83.54 | - | - |
| | MACIL-SD$_{MM\ '22}$ [53] | - | I3D+VGGish | 83.40 | - | - |
| | MGFN$_{AAAI\ '23}$ [6] | - | VideoSwin | 80.11 | - | 86.67 |
| | UR-DMU$_{AAAI\ '23}$ [64] | - | I3D RGB | 81.66 | - | **86.97** |
| | CU-Net$_{CVPR\ '23}$ [57] | - | I3D+VGGish | 81.43 | - | 86.22 |
| | CoMo$_{CVPR\ '23}$ [7] | - | I3D RGB | 81.30 | **97.60** | 86.10 |
| | PEL4VAD$_{TIP\ '24}$ [29] | ✓ | I3D RGB | 85.59 | 98.14 | 86.36 |
| | VadCLIP$_{AAAI\ '24}$ [47] | ✓ | CLIP | 84.51 | - | 88.02 |
| | HLGAtt$_{CVPR\ '24}$ [11] | - | I3D+VGGish | **86.34** | - | - |
| | TPWNG$_{CVPR\ '24}$ [49] | ✓ | CLIP | 83.68 | - | 87.79 |
| | RTFM$^*_{ICCV\ 21'}$ [37] | - | I3D RGB | 77.37 | 94.32 | 82.80 |
| | MGFN$^*_{AAAI\ 23'}$ [6] | - | I3D RGB | 76.10 | 88.67 | 83.21 |
| | UR-DMU$^*_{AAAI\ 23'}$ [64] | - | I3D RGB | 82.58 | 90.51 | 86.38 |
| Frame- Supervised | RTFM$^*_{ICCV\ 21'}$ [37] | - | I3D RGB | 82.31 | 97.69 | 85.60 |
| | MGFN$^*_{AAAI\ 23'}$ [6] | - | I3D RGB | 81.27 | 94.52 | 85.23 |
| | UR-DMU$^*_{AAAI\ 23'}$ [64] | - | I3D RGB | 86.30 | 95.38 | 88.17 |
| | **Ours** | - | I3D RGB | 88.09 | **98.41**(+0.81) | 89.86 |
| | **Ours** | - | I3D+CLIP | **89.56**(+3.22) | 98.32 | **90.23**(+3.26) |

The symbol "*" denote these methods are reproduced by the official codes on weakly-supervised and frame-supervised setting, respectively.

position of annotated frames with respect to abnormal video or within abnormal interval, enabled by frame-level annotation from Liu and Ma [20]. For ShanghaiTech, the distribution exhibits a Gaussian-like pattern with peak near the center. For UCF-Crime and XD-Violence, the distribution exhibits a clear tendency for annotations to concentrate towards the former part of abnormal videos. The concentration of annotations in the early part of videos indicates that anomaly cues often emerge early, eliminating the need for full video review and thereby validating the efficiency of our SF-VAD annotation protocol.

## 4.3 Evaluation Metrics

Following existing works [34, 57, 46], we employ the Area Under the Curve (AUC) as the primary evaluation metric for ShanghaiTech and UCF-Crime, and Average Precision (AP) for XD-Violence. Furthermore, False Alarm Rate (FAR) with a threshold value of 0.5 is assessed, following previous works [34, 64]. A lower FAR signifies a reduced occurrence of false positives, which is essential for practical applications as it directly contributes to the trustworthiness of detection results.

## 4.4 Performance Comparisons

We conduct comprehensive performance comparisons against state-of-the-art Semi-VAD [35, 12, 26] and Weakly-VAD [34, 10, 27, 29, 11, 50] methods, including recent text-enhanced approaches [29, 47, 49]. As depicted in Table 1, our proposed method consistently achieves state-of-the-art performance, notably maintaining its effectiveness even when using only I3D [3] features. On the large-scale XD-Violence, our method achieves a superior AP of 89.56%, representing a substantial 3.22% abso-

Table 2: FAR comparison with state-of-the-art methods.

| Supervision | Methods | Text | Feature | XD(%) | SH(%) | UCF(%) |
|---|---|---|---|---|---|---|
| Semi-Supervised | Conv-AE$_{CVPR\ '16}$ [12] | - | - | - | - | 27.2 |
| | GODS$_{ICCV\ '19}$ [42] | - | BoW+TCN | - | - | 2.10 |
| Weakly-Supervised | MIL-Rank$_{CVPR\ '18}$ [34] | - | C3D RGB | - | 0.15 | 1.90 |
| | GCN$_{CVPR\ '19}$ [63] | - | TSN RGB | - | - | 0.10 |
| | AR-Net$_{ICME\ '20}$ [41] | - | I3D RGB | - | 0.10 | - |
| | MIST$_{CVPR\ '21}$ [10] | - | I3D RGB | - | 0.05 | 0.13 |
| | CRFD$_{TIP\ '21}$ [45] | - | I3D RGB | - | - | 0.72 |
| | UR-DMU$_{AAAI\ '23}$ [64] | - | I3D RGB | 0.65 | - | - |
| | PEL4VAD$_{TIP\ '24}$ [29] | ✓ | I3D RGB | 0.75 | 0.00 | 0.43 |
| Frame-Supervised | **Ours** | - | I3D RGB | **0.35** | 0.00 | 0.01 |
| | **Ours** | - | I3D+CLIP | 0.37 | **0.00** | **0.01** |

A lower FAR indicates more reliable anomaly detection.

lute improvement over the best non-text Weakly-VAD methods and outperforming text-augmented approaches. Similarly, on the challenging UCF-Crime dataset with diverse real-world anomaly scenarios, we achieve an AUC of 90.23%, a 3.26% absolute gain over prior SOTA methods. The performance improvement demonstrates the effectiveness of FPL which enables reliable generalization of single frame annotation to intact abnormal temporal scope. As further illustrated in Figure 4, our method outperforms existing approaches [22, 37, 7] across various abnormal classes, especially on visually subtle anomaly types such as Arson, Assault, and Shooting. This improvement is attributed to the precise anomaly reference by SF-VAD, which effectively accentuates abnormal events in their context. In addition, we compare the False Alarm Rate (FAR) of our method with state-of-the-art Semi-VAD [12, 42] and Weakly-VAD [34, 10, 64] baselines, as in Table 2. Our method reduces FAR significantly across multiple datasets, demonstrating its superior ability to suppress pseudo anomalies and distinguish normal patterns within abnormal videos via relevance-aware pattern decoupling, ultimately leading to more reliable anomaly detection.

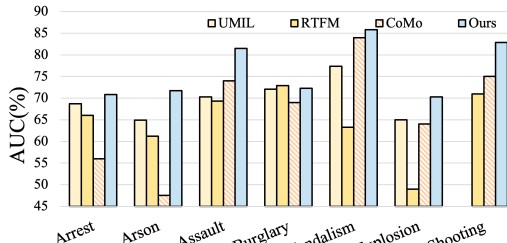

Figure 4: AUC w.r.t. classes on UCF-Crime.

Table 3: Ablation studies on UCF-Crime.

| Baseline | SF | FARE | AEM | ND | AUC(%) | FAR (%) |
|---|---|---|---|---|---|---|
| ✓ | - | - | - | - | 83.67 | 0.62 |
| ✓ | ✓ | - | - | - | 84.36 | 0.59 |
| ✓ | ✓ | ✓ | ✓ | - | 88.63 | 0.41 |
| ✓ | ✓ | ✓ | ✓ | ✓ | 90.23 | 0.01 |

## 4.5 Ablation Studies

To validate the effectiveness of the proposed modules, we conduct comprehensive ablation studies on the UCF-Crime dataset, as summarized in Table 3. The baseline model adopts a Multiple Instance Learning (MIL) strategy under weak supervision. By introducing single-frame supervision into the MIL framework, the model benefits from clearer training signals, resulting in more accurate anomaly detection. Building on this foundation, the integration of FARE and AEM further enhances performance by extending the limited single-frame supervision to a broader temporal scope, allowing the model to capture abnormal patterns more comprehensively. Notably, incorporating the Normal Decoupling (ND) procedure in Stage II leads to substantial performance gains. ND explicitly separates normal contexts from abnormal ones, which significantly reduces the false alarms and further improves the precision and reliability of detection. Additionally, we perform ablation studies on abnormal videos with varying numbers of abnormal events, as in Figure 5. The results show that our method consistently outperforms the baseline, demonstrating the generalization ability of the proposed FPL framework in extending single-frame supervision to discrete abnormal events.

In addition, we conduct ablation studies on the abnormal event mining module and the normal behavior decoupling strategy to further investigate the contribution of each component, as summarized in Table 4. The results show that incorporating key frame selection improves anomaly modeling effectively by extracting more representative frames, while interval mining generally enhances performance by providing richer temporal context. Moreover, decoupling normal behavior from pre-event and post-event contexts offers additional gains, where each part contributes modest improvements and their combination achieves the best overall performance, 90.23% AUC, suggesting their complementary effects in reducing pseudo anomalies and enhancing feature discrimination.

Table 4: Ablation study of the abnormal event mining algorithm and normal decoupling strategy.

| Abnormal Event Mining | | UCF(%) | Normal Decoupling | | UCF(%) |
|---|---|---|---|---|---|
| Key Frame | Interval Mining | | Pre-event | Post-event | |
| - | - | 85.13 | - | - | 88.63 |
| ✓ | - | 86.69 | ✓ | - | 89.83 |
| - | ✓ | 88.82 | - | ✓ | 89.05 |
| ✓ | ✓ | 90.23 | ✓ | ✓ | 90.23 |

## 4.6 Qualitative Results

To substantiate the effect of our method intuitively, the predicted anomaly scores of hard cases are visualized on the challenging UCF-Crime, compared to SOTA methods [6, 64, 29]. As illustrated in Figure 6a, our method can accurately detect varied-length abnormal intervals while precisely distinguishing the subtle normal intervals lying between abnormal intervals. To exhibit the capability of the proposed method in separating abnormal and normal features for precise anomaly detection, we visualize the feature distribution of intermediate features. As depicted in Figure 6b, abnormal and normal features exhibit a clear separation, with higher anomaly relevance values predominantly associated with abnormal instances. This distinct margin highlights the effectiveness of our frame-guided anomaly relevance estimation and relevance-aware pattern decoupling in isolating abnormal cues from normal contexts. Additional qualitative results are presented in the Appendix J.

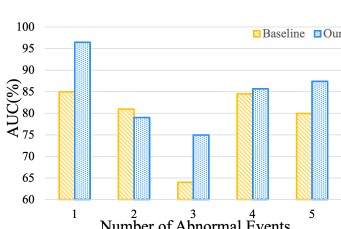

Figure 5: Ablations w.r.t. number of abnormal events.

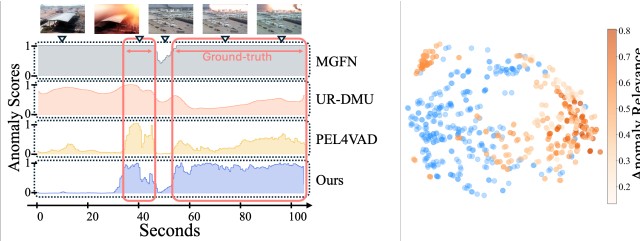

(a) Anomaly Scores.  (b) Feature distribution.

Figure 6: Qualitative results on UCF-Crime.

## 5 Conclusion

In conclusion, we introduce single-frame supervised VAD, a novel paradigm that offers favorable annotation efficiency and precise anomaly reference by a single annotated abnormal frame per abnormal video. Correspondingly, the proposed frame-guided progressive learning effectively generalizes sparse supervision toward robust event-level anomaly understanding. Extensive experiments on the proposed SF-VAD benchmarks demonstrate that our method consistently achieves superior detection performance across varying numbers of anomaly events and diverse anomaly types. These results validate SF-VAD's capability to accurately detect complex and subtle anomalies, paving the way for more practical and scalable video anomaly detection paradigms.

## Acknowledgement

This work was supported by the National Nature Science Foundation of China (62322211), the "Pioneer" and "Leading Goose" R&D Program of Zhejiang Province (2024C01023).

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

# A  Table of Contents

The appendix is organized as follows:

- Section B describes the construction process of the single-frame supervised VAD datasets.
- Section C provides additional dataset statistics.
- Section D provides annotation time estimation of different VAD paradigms.
- Section E provides detailed descriptions of the datasets.
- Section F outlines the baseline architecture.
- Section G presents the implementation details.
- Section H provides further ablation studies.
- Section I provides hyperparameter analysis.
- Section J showcases additional qualitative results.
- Section K discusses the difference between our method and related works.
- Section L discusses the limitation and further work.

# B  Dataset Construction

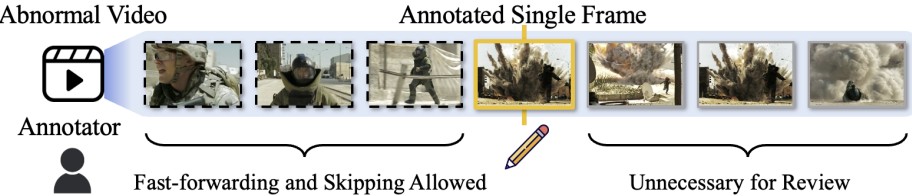

Figure 7: Illustration of single frame annotation procedure.

To adapt single-frame supervision for video anomaly detection (VAD), one of the primary challenges is the absence of appropriate datasets annotated with fine-grained frame-level labels. Most existing VAD benchmarks are formulated into semi-supervised and weakly-supervised settings, where only video-level ground-truth is provided, which falls short of the granularity required for frame-level supervision. Although Liu and Ma [20] offer frame-level annotations for the training set of UCF-Crime [34], the quality of annotation is sub-optimal, with omission of abnormal events and inexact localization of event boundaries, limiting the effectiveness for sampling frame-supervision from full annotations, as is commonly done in tasks, e.g., moment retrieval [9].

To address this limitation, we construct three high-quality, human-annotated Single-Frame supervised VAD (SF-VAD) datasets based on publicly available VAD benchmarks: ShanghaiTech Campus [21], UCF-Crime [34], and XD-Violence [46]. To maximize annotation efficiency while ensuring labeling accuracy, our SF-VAD datasets follow a practical single-frame annotation protocol that reflects how annotators behave in realistic labeling scenarios. Thereby, the constructed datasets not only enable the study of SF-VAD under realistic supervision constraints, but also reveal genuine human annotation preferences, offering valuable insights for developing methods that adapt to such real-world biases. Moreover, the protocol provides a scalable and cost-effective foundation for constructing large-scale SF-VAD benchmarks in the future.

Specifically, these SF-VAD datasets are annotated through a carefully designed crowdsourced annotation process where twelve human annotators participate. Before starting, annotators had to familiarize themselves with the definitions of various abnormal behaviors, such as abuse, riot, and shoplifting, and then pass a preliminary annotation test to ensure the annotation accuracy. Each annotator works independently, and cross-validation is conducted to ensure the consistency and quality of the annotations. As depicted in Figure 7, to streamline the annotation process and ensure high accuracy, we provide annotators with the following guidelines: 1) Annotators are permitted to freely navigate the video timeline (e.g., via fast-forwarding or skipping) to identify potential abnormal events efficiently. 2) Annotators shall label exactly one frame per video, selected only when they are fully confident that the frame corresponds to an abnormal event. Once all individual annotations are complete, a cross-verification process is performed to identify inconsistencies. Discrepancies

between annotations are reviewed and corrected, ensuring the final annotations accurately reflect the frames where abnormal events occur.

## C  Dataset Statistics

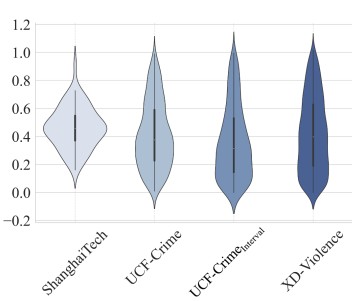

Figure 8: Violin plot of relative position of annotated frames. UCF-Crime$_{\text{Interval}}$ refers to relative position within abnormal events, while other entries indicate relative position w.r.t abnormal videos.

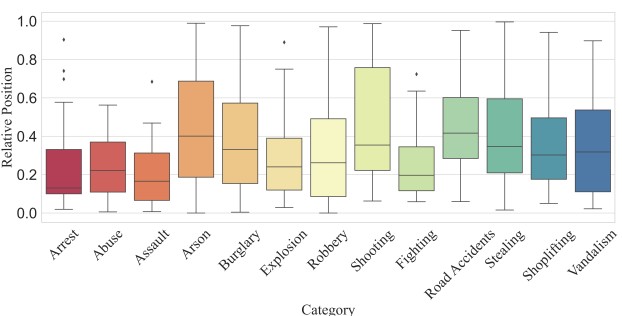

Figure 9: Box plot of the relative position of annotated single frames in UCF-Crime w.r.t different anomaly classes.

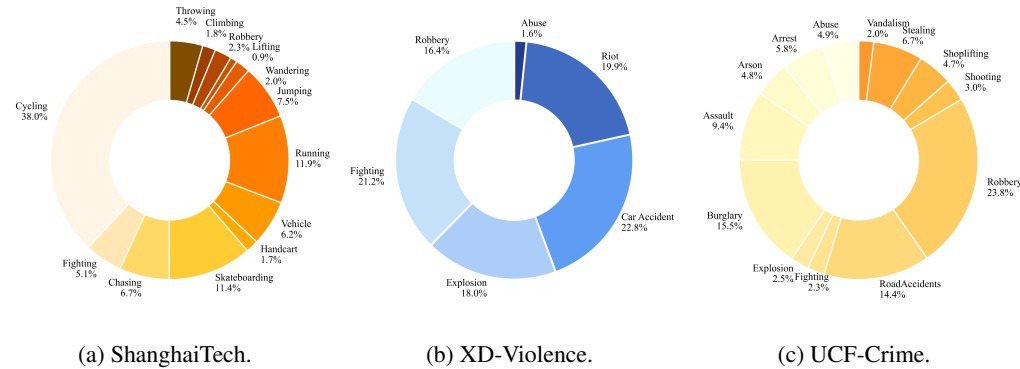

(a) ShanghaiTech.    (b) XD-Violence.    (c) UCF-Crime.

Figure 10: Proportion of total abnormal video duration accounted for each abnormal category across three VAD datasets.

This section provides a more detailed analysis of the SF-VAD dataset statistics, offering insights into the characteristics of the annotated frames. First, we illustrated the relative position of annotated frames within abnormal intervals and videos in Figure 8. The distribution of annotated frames within the ShanghaiTech dataset exhibits a near-Gaussian distribution, with its peak centered around the middle of the video. This suggests that abnormal videos in ShanghaiTech tend to comprise clear pre-event, abnormal event, and post-event stages, with the abnormal events typically unfolding near the temporal center of the videos. For both UCF-Crime and XD-Violence datasets, the annotated frames are predominantly concentrated towards the earlier segments of the videos. This bias implies that initial frames in these datasets often contain critical cues indicative of an impending or ongoing anomaly, which also potentially leads to significant reductions in annotation time.

Furthermore, we visualize the relative position of annotated frames within anomalous events for various anomaly classes in the UCF-Crime dataset, as depicted in Figure 9. Generally, the majority of annotated frames across all anomaly classes are indeed skewed towards the beginning of the video. Notably, for classes such as 'Abuse,' 'Assault,' and 'Fighting,' which typically involve rapid and drastic movements, the variance in the relative position of annotated frames is remarkably small. This concentrated annotation suggests that the critical distinguishing features for these events are often visually salient and emerge early in the temporal sequence. This observation also substantiates

SF-VAD's efficiency, as it can direct annotators to these crucial early frames, thereby streamlining the annotation process without full video review.

Beyond the temporal distribution, we also analyze the proportion of total abnormal video duration accounted for by each abnormal category within the training sets of each dataset. As shown in Figure 10, for UCF-Crime, certain anomaly classes, e.g., 'Vandalism' and 'Shooting', constitute a relatively minor proportion of the overall training data. Despite this limited representation, our SF-VAD method achieves refined detection results for these underrepresented classes, as shown in Section 4.4. This remarkable performance on 'trivial' or low-shot classes underscores the effectiveness of SF-VAD in providing fine-grained guidance to highlight subtle anomalies from the context. By leveraging the limited yet informative cues, SF-VAD demonstrates its capability to learn robust representations even from sparse data, which is a significant advantage in real-world anomaly detection scenarios where certain anomalies are inherently rare.

## D    Annotation Time Estimation

In practice, data annotation is a highly intricate process that encompasses not only the explicit time required for watching videos and assigning labels, but also a significant amount of additional effort that is often overlooked. This includes reviewing and replaying video segments to identify specific frames, verifying the temporal boundaries of anomalous events, rechecking annotations for consistency, conducting cross-validation, resolving annotation conflicts, and training annotators. Given the diverse and layered nature of these activities, accurately measuring the true annotation time becomes exceedingly difficult. Therefore, in this work, we estimate the annotation cost using a theoretical lower bound based on a set of practical assumptions. The annotation time versus detection performance is depicted in Figure1c in the main paper.

**Fully-supervised VAD** utilizes frame-level labels, which requires annotators to watch all videos from beginning to end at least once. Accordingly, the theoretical lower bound of annotation time is equivalent to the total duration of the dataset. In practice, however, the actual annotation cost is significantly higher due to the exhaustive temporal localization of abnormal event boundaries, which often necessitates frequent playback, meticulous inspection, and multiple rounds of verification to ensure temporal accuracy and consistency.

**Semi-supervised VAD** leverage normal videos only, however, the annotators need to watch the entire video snippets to make sure that the videos do not contain anomalies of any form. Therefore, the lower bound of annotation time equals the total duration of the normal videos in the dataset.

**Weakly-supervised VAD** uses video-level binary labels. For videos in the test set, the estimated annotation time is equivalent to the total duration of the test videos. For normal videos in the train set, the estimated annotation time equals the total duration as well, since the annotators need to watch the entire video to make sure it is a normal one. For abnormal videos in the training set, the estimated annotation time is estimated as the sum of time an annotator spends observing an abnormal frame within an abnormal video.

**Single-Frame supervised VAD** leverages single-frame annotation. Assuming that we elaborately devise an annotation platform, that enables the annotators to label the abnormal frame as soon as they identify one and let annotation proceed, the low bound annotation time is equal to weakly-supervised VAD. Notably, in piratical scenarios, the annotation time of single frame supervised VAD is slightly larger than weakly-supervised VAD, since single frame annotations involve playback from short anomalies and extra cross validation time to handle the conflict of annotations.

## E    Dataset Description

In this work, we construct SF-VAD benchmarks based on three widely-applied VAD datasets, ShanghaiTech Campus [21], UCF-Crime [34], XD-Violence [46], which cover broad range of abnormal behaviors, scene types, lengths and frequencies of abnormal events, and varying camera perspectives, as depicted in Table 5. The examples of annotated abnormal frames across various anomaly classes is depicted in Figure 11.

**ShanghaiTech Campus** [21] comprises 437 videos from 13 fixed-view campus surveillance cameras. The abnormal types are cycling, chasing, cart, fighting, skateboarding, vehicle, running, jumping,

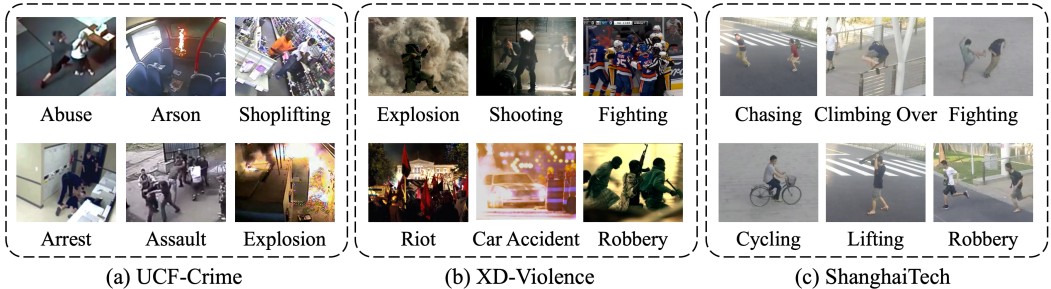

Figure 11: Illustrative examples of annotated abnormal frames across various anomaly classes.

wandering, lifting, robbery, climbing over, throwing. The background of the frames is rather steady and contains less noise, which highlights the behaviors within the frames.

**UCF-Crime** [34] comprises 1900 videos collected from a variety of sources including videos from surveillance cameras and social media with a total duration of 128 hours. The dataset covers 13 real-world anomalies of crimes including abuse, arrest, arson, assault, burglary, explosion, fighting, road accident, shooting, shoplifting, stealing, vandalism and robbery. The representations of the anomalies are varied and differentiated which increases the challenge of the detection by requiring a more thorough understanding of the anomaly semantics.

**XD-Violence** [46] is the largest and most challenging multi-modal VAD dataset containing 4754 untrimmed videos with a total duration of 217 hours. The dataset contains videos from various sources such as movies, social media, car cameras, surveillance, and games where exist extensive artistic expressions such as changing perspective, view zooming, dynamic lighting, and rapid camera movements. The above characteristics of the datasets draw non-negligible difficulty to anomaly detection models. It covers anomalies of 7 types including abuse, car accidents, explosions, fighting, riots, robbery and shooting.

Table 5: Comparison of video anomaly detection datasets.

| Dataset | Domin | #Videos | #Train Abn. | #Train Nor. | #Test Abn. | #Test Nor. | #Abn. Types | Resolution |
|---|---|---|---|---|---|---|---|---|
| ShanghaiTech [21] | Campus | 437 | 63 | 175 | 44 | 155 | 13 | 856×480 |
| UCF-Crime [34] | Crime | 1900 | 810 | 800 | 140 | 150 | 13 | Multiple |
| XD-Violence [46] | Violence | 4754 | 1905 | 2049 | 500 | 300 | 7 | 640×360 |

# F    Baseline

The architecture of the overall framework is depicted in Figure2 in the main paper. Concretely, given an untrimmed video, pertained feature encoders [30, 3] are employed to obtain multi-modal features. Subsequently, the features are passed through the Transformer-based Temporal Modeling (TTM) module and detector to predict frame-level anomaly scores. Building upon the temporal features, multi-layer convolutional networks are employed as evidence encoder and anomaly detector to predict evidence for relevance estimation and the final anomaly scores.

Considering the trade-off of computational overhead and detection performance, the input videos are split into 16-frame non-overlapping clips. Pre-trained frozen encoders are utilized to extract embedding features, formulating clip feature sequences. Embedding features are denoted as $x \in \mathbb{R}^{N \times D_m}$ where $N$ equals the number of the clips and $D_m$ is the dimension of the features.

Owing to resounding success in natural language processing areas, Transformer [40] has been verified as a highly effective architecture for capturing global dependencies. And it has been successfully employed in temporal modeling [52, 1]. Therefore, we apply TTM module, following [29], to capture multi-scale temporal cues for evidence and anomaly score prediction, as depicted in Figure 12.

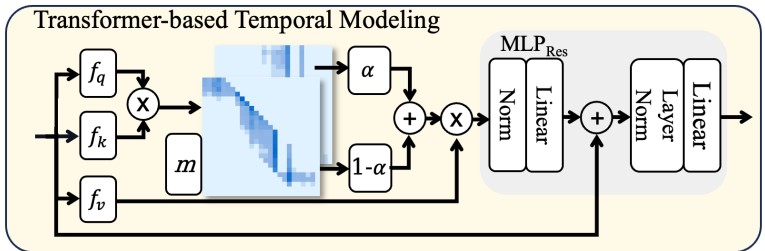

Figure 12: Architecture of Transformer-based Temporal Modeling module.

First, the attention mechanism's similarity matrix $\boldsymbol{m} \in \mathbb{R}^{N \times N}$ is computed with dynamic position encoding $\mathcal{E} \in \mathbb{R}^{N \times N}$ added to incorporate temporal position prior:

$$
\begin{aligned}
\boldsymbol{m} &= f_q(\boldsymbol{x}) \cdot f_k(\boldsymbol{x})^\top + \mathcal{E}, \\
\mathcal{E}_{j,k} &= \exp\left(-\left|\gamma(j-k)^2 + \beta\right|\right),
\end{aligned}
\tag{13}
$$

where $f(\cdot)$ refers to linear layers and $j, k \in [1, N]$ indicate index of clips. $\gamma$ and $\beta$ represent learnable weight and bias. Then, global attention feature $\boldsymbol{f} \in \mathbb{R}^{N \times D_h}$ is computed based on the similarity matrix and the linear projection of $\boldsymbol{x}$. The process can be denoted as follows:

$$
\boldsymbol{f} = \mathrm{softmax}\left(\frac{\boldsymbol{m}}{\sqrt{D_h}}\right) \cdot f_v(\boldsymbol{x}),
\tag{14}
$$

where $D_h$ indicates the hidden dimension. To highlight short-range temporal attention of events and solve long-range noise, the similarity matrix is masked by a sliding window. The process can be denoted as:

$$
\widetilde{\boldsymbol{m}}_{ij} = \begin{cases} \boldsymbol{m}_{ij}, & j \in \left[\max\left(0, i - \left\lfloor \frac{w}{2} \right\rfloor\right), \min\left(i + \left\lfloor \frac{w}{2} \right\rfloor, N\right)\right] \\ -\infty, & \text{otherwise} \end{cases},
\tag{15}
$$

where $w$ refers to the window size and $\widetilde{\boldsymbol{m}}$ indicates local similarity matrix. Correspondingly, local attention feature $\widetilde{\boldsymbol{f}} \in \mathbb{R}^{L \times D_h}$ is computed by Equation 14. Then, global and local features are fused by gate weight $\alpha$. Subsequently, a residual connection is utilized followed by layer normalization to derive temporal feature $\boldsymbol{f}^t \in \mathbb{R}^{L \times D_m}$, which can be formulated as:

$$
\begin{aligned}
\boldsymbol{f}^t &= f_o\left(\mathrm{Norm}\left(\alpha \cdot \boldsymbol{f} + (1 - \alpha) \cdot \widetilde{\boldsymbol{f}}\right)\right), \\
\boldsymbol{z} &= \mathrm{LayerNorm}\left(\boldsymbol{x} + \boldsymbol{f}^t\right),
\end{aligned}
\tag{16}
$$

where $\mathrm{Norm}(\cdot)$ denotes a composite of power normalization [54] and L2 normalization. Eventually, TTM acquires multi-scale temporal feature $\boldsymbol{z} \in \mathbb{R}^{N \times D_m}$. Eventually, multi-layer convolutional networks are employed as evidence encoder and anomaly detector to predict evidence $\boldsymbol{e} \in \mathbb{R}^{N \times 2}$ for relevance estimation, which can be denoted as:

$$
\begin{aligned}
\mathrm{MLP} &= \mathrm{Dropout}\left(\mathrm{GELU}(\mathrm{Conv}(\cdot))\right), \\
\boldsymbol{e} &= \mathrm{LeakyReLU}\left(f_t\left(\mathrm{MLP}\left(\mathrm{MLP}\left(\boldsymbol{z}\right)\right)\right)\right),
\end{aligned}
\tag{17}
$$

where $\mathrm{Conv}(\cdot)$ refers to one-dimension convolution followed by GELU [13] and $f_t(\cdot)$ represents causal convolutional layer. LeakyReLU corresponds to the activation function [? ]. Similarly, the final anomaly scores $\hat{\boldsymbol{y}} \in \mathbb{R}^N$ can be predicted as:

$$
\hat{\boldsymbol{y}} = \sigma\left(f_t\left(\mathrm{MLP}\left(\mathrm{MLP}\left(\boldsymbol{z}\right)\right)\right)\right),
\tag{18}
$$

where $\sigma$ indicates the sigmoid activation function.

## G  Implementation Details

**Feature Extraction.** To extract video features, we follow existing methods [44, 29, 46]. We apply the I3D [3] video encoder that is pre-trained on Kinetics [15] dataset, to acquire video motion features. I3D processes each video frame and aggregates temporal context over a sequence of frames,

enabling it to extract rich, motion-aware features from the video. Video features are extracted from *global_pool* layer from the I3D encoder which is 1024 dimensions. To acquire video appearance features, we utilize CLIP [30](ViT-B/16) image encoder. CLIP extracts visual semantic features for each frame that generally focus on the overall appearance. The acquired appearance features contain 512 dimensions. For the trade-off of detection performance and computational overhead, each video is split into 16-frame non-overlapping clips. Notably, we employ a crop augmentation strategy to enhance the generalization ability. For UCF-Crime and ShanghaiTech datasets, we apply a ten-crop augmentation strategy, which includes crops from the center, four corners, and their mirrored counterparts. For XD-Violence dataset, we employ a five-crop augmentation strategy, which includes crops from the center and four corners.

**Hyperparameter.** The hidden dimension $D_h$ of transformer-based temporal modeling module is set to 128. The initial gate weight $\alpha$ of transformer-based temporal modeling module is set to 0.5. The window size $w$ is set to 5, 9, and 9 for ShanghaiTech, UCF-Crime, and XD-Violence, respectively. The kernel size and stride of the one-dimensional convolutional layer $f_t$ are set to 3 and 1, respectively. In abnormal event mining algorithm, the threshold $\theta_1$ that filters the total variance of similarity is set to 0.1. The threshold $\theta_2$ that controls the prominence of similarity of key frames is set to 0.96. The threshold $\theta_3$ that controls the gap of abnormal events is set to 0.2.

**Training Details.** All experiments are conducted on a single NVIDIA RTX 3090 GPU using PyTorch. During training, the model parameters are initialized by Xavier initialization. The batch size is set to 128. The learning rate is $5 \times 10^{-4}$ initially and controlled by a cosine decay strategy. The parameters are optimized using Adam optimizer. The number of training epochs is set to 50. For the balance between computational overhead and detection performance, the maximum sampling sequence length is set to 200 during the training phase.

## H    Further Ablation Studies

To further validate the effectiveness of our proposed framework, we conduct extensive ablation studies on the UCF-Crime dataset. In Table 6, we study the impact of different supervision paradigms. While the baseline under complete weak supervision only achieves 83.67% in terms of AUC. Gradually increasing the ratio of single-frame supervised training video leads to substantial performance improvements. The hybrid setting with 50% weakly-supervised and 50% single-frame annotations achieves 87.79% AUC. Remarkably, the fully single-frame supervised version reaches 90.23% AUC, demonstrating that concise but precise single-frame supervision is highly effective for anomaly localization. These results suggest that, with comparable annotation cost to weak supervision, single-frame supervision offers a more cost-effective solution by providing fine-grained anomaly cues that substantially improve anomaly localization performance.

Table 6: Ablation study of the ratio of training data under different supervision paradigms.

| Paradigm | Weakly-supervised | Single-frame supervised | UCF |
|---|---|---|---|
| Weakly-supervised | 100% | 0% | 83.67 |
| Hybrid | 75% | 25% | 85.36 |
| | 50% | 50% | 87.79 |
| | 25% | 75% | 88.51 |
| Single-frame supervised | 0% | 100 % | 90.23 |

## I    Hyperparameter Analysis

**Effect of Threshold $\theta_2$.** In UCF-Crime and XD-Violence, we conduct a hyperparameter analysis to investigate the effect of the abnormal event mining threshold $\theta_2$, which controls the required prominence of feature similarity among selected key frames. A larger $\theta_2$ enforces stricter similarity constraints, leading to the selection of more confidently abnormal frames. Conversely, a smaller $\theta_2$ allows for more diverse but potentially noisier frames to be included. As shown in Figure 13a, performance initially improves as $\theta_2$ increases, benefiting from more precise supervision signals. However, overly large values of $\theta_2$ may result in overly conservative frame selection, missing important abnormal cues and leading to performance degradation. Empirically, $\theta_2 = 0.95$ achieves the best performance, striking a good balance between precision and coverage in selected key frames.

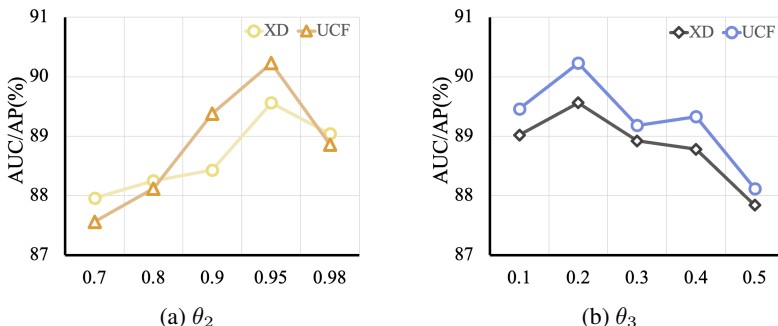

(a) $\theta_2$          (b) $\theta_3$

Figure 13: Hyperparameter analysis of $\theta_2$ and $\theta_3$ in abnormal event mining algorithm in XD-Violence and UCF-Crime.

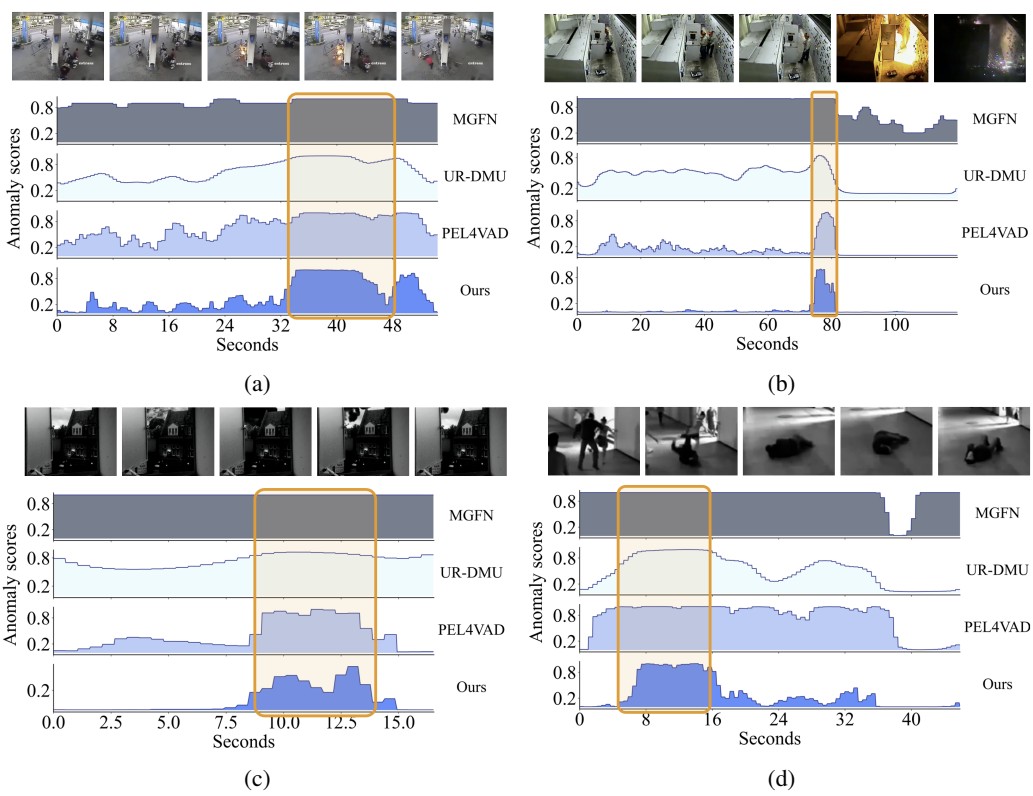

Figure 14: Visualization of anomaly scores in the UCF-Crime dataset. The Y-axis displays the anomaly scores, with 1 indicating abnormal and 0 indicating normal, while the X-axis shows the duration of the videos. The orange-shaded regions highlight the frames where anomalies occur. The frames above are snapshots from the videos. From top to bottom, the anomaly scores are generated by MGFN [6], UR-DMU [64], PEL4VAD [29], and Ours, respectively.

**Effect of Threshold $\theta_3$.** We further analyze the effect of threshold $\theta_3$, which controls the minimum temporal distance between selected abnormal key frames, thereby encouraging diversity among discovered abnormal events. A larger $\theta_3$ enforces broader temporal separation, promoting exploration of distinct abnormal segments. As shown in Figure 13b, moderate values of $\theta_3$ improve performance by preventing supervision collapse into a single event, while overly large values may overlook densely occurring anomalies. As the results show, setting $\theta_3 = 0.2$ yields the best performance, effectively balancing redundancy reduction and anomaly coverage.

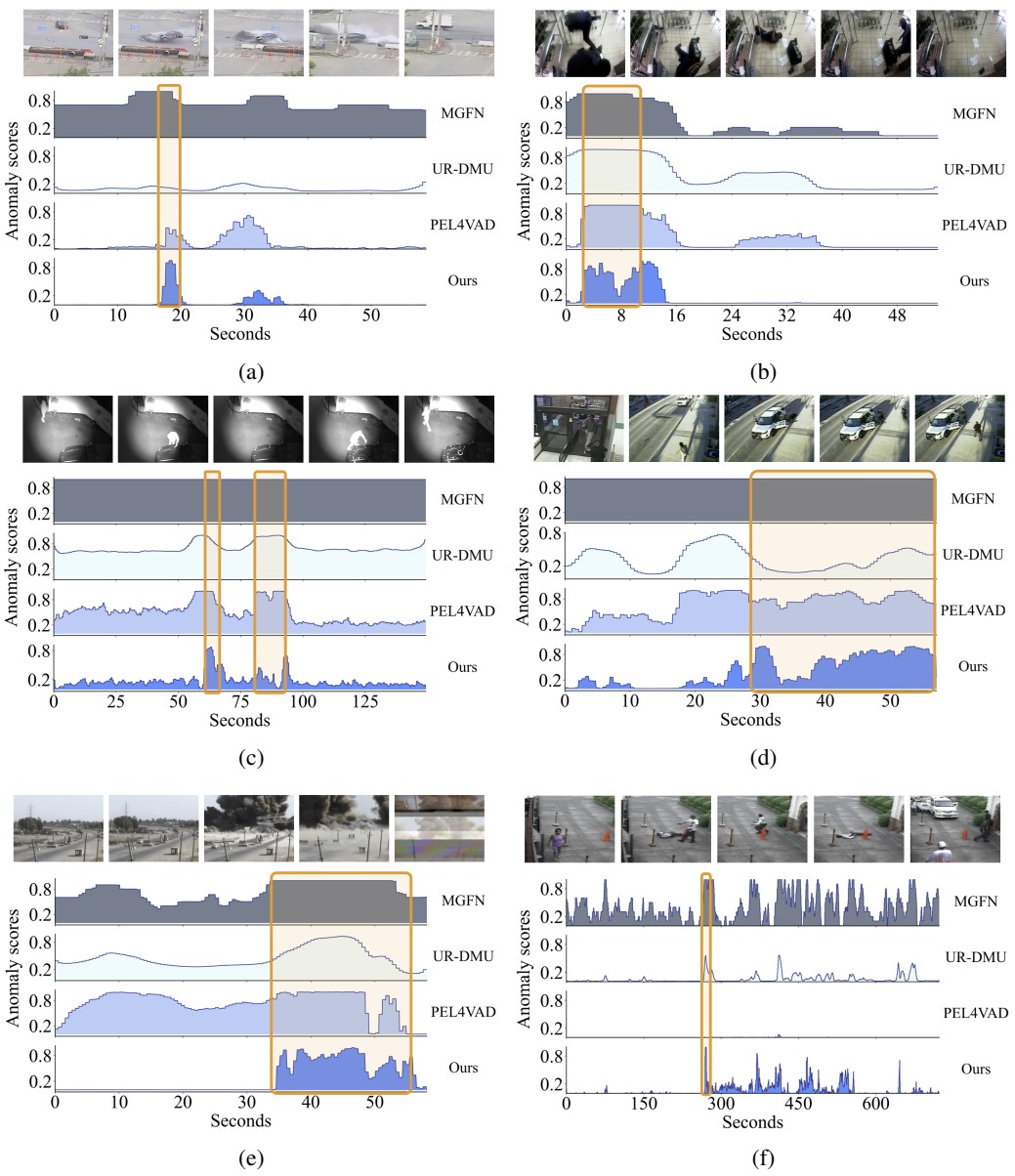

Figure 15: Visualization of anomaly scores in the UCF-Crime dataset. The Y-axis displays the anomaly scores, with 1 indicating abnormal and 0 indicating normal, while the X-axis shows the duration of the videos. The orange-shaded regions highlight the frames where anomalies occur. The frames above are snapshots from the videos. From top to bottom, the anomaly scores are generated by MGFN [6], UR-DMU [64], PEL4VAD [29], and Ours, respectively.

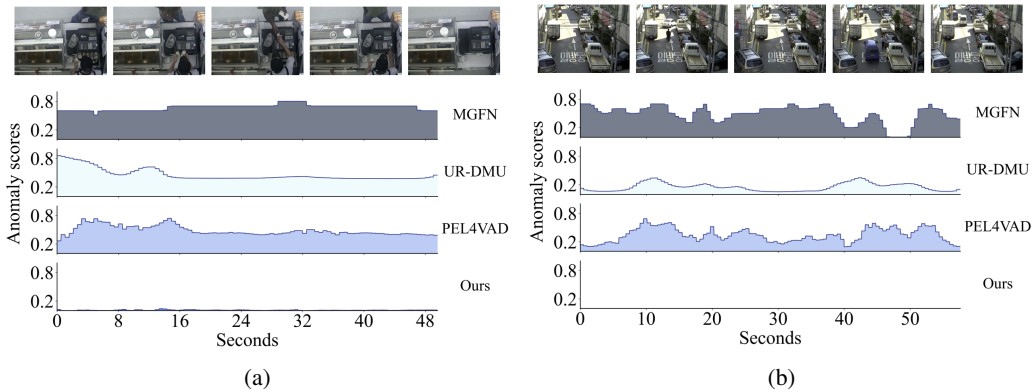

Figure 16: Anomaly scores of normal videos. Smaller anomaly scores indicate fewer false alarms and demonstrate a more reliable detection result. From top to bottom, the anomaly scores are generated by MGFN [6], UR-DMU [64], PEL4VAD [29], and Ours, respectively.

## J    Qualitative Results

To illustrate the effectiveness of our method, we further visualize the anomaly scores of some hard cases with background interference, noisy scenes, subtle abnormal behaviours, and varied anomaly durations, compared with MGFN [6], UR-DMU [64], and PEL4VAD [29].

Figure 14a and Figure 14b visualize the detection results on videos with anomalous events occurring at different temporal scales. Our dynamic anomaly event mining algorithm effectively captures anomaly patterns across varying durations by jointly leveraging anomaly relevance and feature similarity. As a result, it achieves robust detection performance across diverse temporal scopes and produces clear and well-aligned event boundaries. Figure 14c and Figure 14d present detection results on grayscale videos, where the anomalous behaviors are visually subtle and corrupted by significant noise. In such challenging settings, our model still accurately identifies the anomaly duration, attributed to the precise guidance provided by single-frame supervision. Unlike weakly-supervised approaches that rely on coarse temporal labels, the fine-grained supervision facilitates robust learning of discriminative features.

Compared with MGFN, UR-DMU, and PEL4VAD, our method demonstrates more precise temporal localization, effectively capturing the onset and end of temporal episodic anomalies, such as car accidents in Figure 15a and shootings in Figure 15b. These events typically occur and vanish rapidly, making them challenging to detect with weak supervision. Our method successfully localizes them without triggering excessive false alarms, benefiting from the proposed pre-event normal decoupling strategy, which disentangles the contextual patterns preceding abnormal events. This decoupling enables the model to distinguish normal fluctuations from truly anomalous changes. In Figure 15c, we observe a well-localized prediction for a sparse anomaly, alongside effective suppression of pseudo anomalies in unrelated regions. Figure 15d and Figure 15e show ideal detection results on longer anomalous intervals, while Figure 15f demonstrates the ability to detect short anomalies embedded within long abnormal periods. These results highlight our model's ability to decouple fine-scale anomalies from extended contextual sequences, significantly reducing false alarms.

Figure 16 shows detection results on normal videos with high visual similarity to anomalous cases, such as cashier scenes (visually similar to robberies) and traffic scenarios (resembling accidents). Our method yields nearly flat anomaly scores across the entire video, indicating strong confidence in normality. This performance benefits from the proposed pattern decoupling strategy, which explicitly separates abnormal patterns from high-frequency but non-anomalous behaviors. Unlike prior methods that often confuse visually similar contexts, our model learns semantically meaningful representations that generalize well to hard negatives, enabling accurate rejection of false positives in visually ambiguous settings.

# K    Discussion of Related Methods

This section discusses the difference between our method with related works, including the supervision paradigm [59] and methods [65, 36].

Recently, Zhang et al. [59] study glance annotation in VAD, leveraging a frame annotation per abnormal event. Since multiple abnormal events may be involved in an abnormal video, such glance annotation typically requires *multiple* frame annotations per abnormal video, which imposes a high demand on the comprehensiveness of the labeling. On the one hand, this labeling process is more labor-intensive, as a full video review is necessary to ensure the completeness of the annotations. On the other hand, glance annotation requires precise temporal localization of abnormal events, as annotators must label each frame within distinct abnormal events, which necessitates validating both the onset and the conclusion of these events. As a result, the theoretic low bound of annotation time of glance supervision is close to that of fully supervision. In contrast, as depicted in Section C, our single-frame supervised paradigm increases annotation efficiency dramatically compared to glance annotation, as full video review and exhaustive temporal localization are not required in SF-VAD. As a consequence, the theoretic annotation time of single frame supervision is closed to weak supervision.

From a methodological perspective, glanceVAD [59] integrates UR-DMU [64] framework with temporal Gaussian splatting to identify static abnormal intervals, where the variance of Gaussian distribution is static as hyperparameter setting. In contrast, our Frame-guided Progressive Learning (FPL) takes anomaly relevance and feature similarity into consideration to dynamically prob the abnormal event intervals in a reliable way. In addition, FPL decouples normal context in abnormal videos to suppress false alarms, while significantly reducing the annotation burden.

Previous works [65, 36] employ evidential learning to solve VAD problems as well. Zhu et al. [65] integrate evidential learning to select reliable snippets by evidential learning to solve open-set VAD problems. Sun et al. [36] capture the deviation of normal samples as anomalies by evidential learning in semi-supervised VAD paradigm. Fundamentally, our FPL differs from previous methods in the following aspects. First, we leverage evidential learning to estimate anomaly relevance, where only annotated frame is involved to ensure a noise-free anomaly evidence learning process, replacing top-k sampling procedure that introduces noise and destabilizes the training. Second, we leverage Beta distribution in evidential learning instead of Dirichlet distribution for VAD, as a binary classification problem. Third, to encourage relevance learning and predominant evidence, we incorporate regularization term $\mathcal{L}_{\mathrm{KL}}$. As a result, we realize a reliable anomaly relevance estimation by evidential learning.

# L    Limitation and Future Work

In the current approach, the extension from single-frame to multiple anomaly events relies primarily on feature similarity for anomaly detection. While this method shows promising results in the context of the experiments conducted, it places significant demands on the discriminative power of the features. As the complexity of the scenarios increases, the challenge lies in extracting more distinctive features that can effectively differentiate between various anomaly events. Moreover, the ability to reliably explore dynamic, continuous multiple anomalies over time remains an open issue. The model's current formulation may not fully capture the temporal dependencies and interrelations between anomalous segments. Therefore, future work will focus on enhancing feature extraction techniques and developing more robust dynamic strategies to improve the model's capability in detecting multiple anomalies in a continuous sequence.

In addition, current inexact supervision primarily focuses on the temporal dimension, leveraging frame-level annotations for anomaly detection. However, the spatial aspect remains relatively unexplored. A promising direction for future work is to extend this supervision to the spatial domain, where incorporating point-level supervision could highlight the anomalous objects or regions within each frame. By doing so, the model could achieve more accurate spatiotemporal anomaly localization, identifying both the occurrence and the specific spatial location of the anomaly. This would allow for a more granular understanding of abnormal events, further enhancing the model's capability to detect and localize anomalies across both space and time. Therefore, future research will explore methods to integrate spatial cues into the existing framework to improve the robustness and precision of anomaly detection.

