# OpenReview forum: "Generalizing Single-Frame Supervision to Event-Level Understanding for Video Anomaly Detection"
_NeurIPS.cc/2025/Conference — NeurIPS 2025 poster_

### Official Review · Reviewer_mbCU · 2025-07-02

**Clarity:** 3
**Significance:** 3
**Originality:** 3
**Rating:** 4
**Confidence:** 4

**Summary:**

This paper proposes Single-Frame Supervised Video Anomaly Detection (SF-VAD), using one annotated abnormal frame per video to balance annotation efficiency and detection precision. The Frame-guided Progressive Learning (FPL) framework estimates anomaly relevance via evidential learning and generalizes supervision to event-level by mining abnormal intervals and decoupling normal patterns. Three SF-VAD benchmarks are constructed, and experiments show superior AUC and lower false alarm rates compared to prior methods. However, limitations in multi-event robustness and computational analysis weaken its broader impact.

**Questions:**

1.lti-Event Robustness Test: Provide AUC scores for video clips with ≥3 disjoint abnormal events in UCF-Crime to validate FPL’s ability to handle temporal complexity.

2.Computational Efficiency Metrics: Report inference time per video and model parameters compared to baselines (e.g., PEL4VAD, VadCLIP) to clarify trade-offs between performance and cost.

3.Semi-Supervised Baseline Addition: Include results from recent semi-supervised methods (e.g., MULDE, SWAD) to explicitly contrast SF-VAD’s reliance on minimal abnormal labels.

4.Evidential Learning Justification: Add a brief comparison with standard cross-entropy loss in Stage I to explain why evidential learning better captures anomaly uncertainty.

**Ethical Concerns:**

["NO or VERY MINOR ethics concerns only"]

**Final Justification:**

Thank you for the response. After considering the comments from the other reviewers, I think this is an acceptable paper.

**Limitations:**

The authors mention limitations in the supplemental material (e.g., reliance on feature similarity), but they do not adequately discuss how SF-VAD performs when the annotated frame is ambiguous or non-representative of the full anomaly. This oversight raises concerns about the method’s robustness in noisy real-world scenarios.

**Quality:**

2

**Strengths And Weaknesses:**

Strengths:

1.Novel Supervision Paradigm: SF-VAD reduces annotation cost significantly while providing fine-grained anomaly guidance, addressing a critical challenge in VAD.

2.Effective Two-Stage Framework: FPL’s progressive learning from sparse frames to events demonstrates strong generalizability, as shown in ablation studies (Table 3).

3.Dataset Contribution: The manually re-annotated SF-VAD benchmarks offer practical single-frame supervision datasets for future research.

Weaknesses:

1.Limited Multi-Event Analysis: The paper lacks quantitative results on videos with complex, overlapping abnormal events (e.g., UCF-Crime’s “Riot” class), leaving uncertainty about FPL’s scalability.

2.Computational Overhead Omission: No comparison of inference time or model size with lightweight baselines (e.g., Conv-AE) is provided, hindering assessment of practical deployment feasibility.

3.Incomplete Baseline Comparison: Missing direct comparisons with recent semi-supervised methods (e.g., MULDE, CVPR 2024) limits clarity on SF-VAD’s positioning in the VAD landscape.

4.Theoretical Underpinnings: The evidential learning formulation (Sec. 3.3) lacks intuitive justification for its choice over standard probabilistic models, making it harder for readers to contextualize.

---

> ### Author Rebuttal · Authors · 2025-07-31
>
> We deeply appreciate the reviewer's meticulous assessment and profound insights. The positive recognition of our **novel supervision paradigm, effective two-stage framework, and dataset contribution** is especially encouraging. We're also thankful for the constructive criticism, which is invaluable for refining our manuscript. Here are our responses to the concerns:
>
> ---
> **Response to Q1: Multi-event Analysis**
>
> For multi-event generalization validation, class-wise performance on XD-Violence is further evaluated, where more complex behaviors, e.g., _Riot_, and interfering scenarios are involved.
> || Abuse|Explosion|Riot|Fighting|overall|
> |-|-|-|-|-|-|
> |PEL4VAD|70.5|63.9|96.5|83.8|85.26|
> |FPL(Ours)|79.41|68.22|97.63|81.33|89.56|
>
> Notably, our performance steadily surpasses that of the SOTA weakly-supervised methods across diverse abnormal classes, which aligns with the evaluation on UCF-Crime:
> ||Arrest|Arson|Assault|Burglary|Vandalism|Explosion|Shooting|
> |-|-|-|-|-|-|-|-|
> |UMIL|68.7|64.9|70.3|72.1| 77.4|65.0|-|
> |RTFM|66.1|61.2|69.3|**72.9**|63.3|49.5|71.7|
> |CoMo| 56.2| 47.5| 74.1| 69.4| 83.8| 64.6| 75.1|
> |**FPL (Ours)**|**70.8**|**71.7**|**81.5**|72.3|**85.8**|**70.3**| **82.9**|
>
> These results further confirm the generalizability of FPL across varied abnormal behaviors. With the combination of dynamic anomaly relevance estimation and static feature similarity evaluation, diverse abnormal intervals are mined, which ultimately enables accurate capture of the essence of abnormal behaviors even in diverse and noisy scenarios.
>
> ---
> **Response to Q2: Computational Overhead**
>
> For practical usability evaluation, model parameters and inference time are further investigated:
> ||Supervision| Parameters (↓)| Inference Time (↓)|UCF-Crime AUC (↑)|
> |-|-|-|-|-|
> |Conv-AE|Semi-supervised|10.14 M|-|-|
> |MGFN| Weakly-supervised|28.46 M|3.70 s|86.67|
> |RTFM| Weakly-supervised|24.72 M|3.23 s|84.30|
> |UR-DMU|Weakly-supervised| 6.49 M|12.39 s|86.97|
> |VadCLIP|Weakly-supervised (with Text Features) |162.26 M|5.65 s|88.02
> | **FPL(Ours)** | Single-Frame supervised|**2.21 M**|**0.51 s**|**90.23**|
>
> Remarkably, our SF-VAD method demonstrates **a smaller parameter size**, much **faster inference time**, and **better detection performance** than previous state-of-the-art semi-supervised and weakly-supervised VAD methods. The superior performance-cost trade-off stems from the precise abnormal learning target enabled by FPL, which allows a lightweight network architecture to effectively capture diverse anomalous cues, without computationally expensive memory modules and text encoders.
>
> ---
> **Response to Q3: More Complete Baseline Comparison**
>
> Due to the page limitation in initial submission, the semi-supervised methods are not included. As shown in Sec. H in Supp., we demonstrate more comprehensive comparison results with the state-of-the-art semi-supervised methods including, MULDE[1], AED-MAE[2], MGEnet[3], LANP[4], etc. While we cannot locate the specific *SWAD* literature, if the reviewer could kindly provide it, we would be delighted to perform this comparison and include the appropriate citation in the revised manuscript. This would further illustrate our method's broad applicability and effectiveness.
> |Method|Supervision|ShanghaiTech|UCF-Crime|XD-Violence|
> |-|-|-|-|-|
> |MULDE|Semi-supervised|81.3|78.50|-|
> |AED-MAE|Semi-supervised|79.1|-|-|
> |MGEnet|Semi-supervised|86.9|-|-|
> |LANP|Semi-supervised|88.32|80.02|-|
> |VadCLIP|Weakly-supervised (with Text Features)|-|88.02|84.51|
> |**FPL(Ours)**|Single-Frame supervised|98.32|90.23|89.56|
>
> The complete evaluation results will be included in the main manuscript in the revision.
>
> ---
> **Response to Q4: Intuitive Justification for Evidential Learning**
>
> As mentioned in **line 150-152**, feature similarity itself is not fully dependable due to its insensitivity to subtle changes in local spectrum. Therefore, evidential learning is introduced, which **provides uncertainty in its predictions as an estimation of the discrepancy from annotated anomalies**. Specifically,  FPL first employs an evidential learning model to learn noise-free abnormal patterns using the reference of SF-VAD solely. During the subsequent stage, evidential learning model predicts uncertainty to measure the deviation between input distribution and modeled anomaly, which serves as a proxy for anomaly relevance. Via the relevance, we generalize the anomaly supervision from a single frame into more complete scope. For a detailed explanation:
>
> **Why Evidential Learning**: In comparison to standard probabilistic models, evidential learning estimates the deviation from the learnt accurate abnormal patterns by prediction uncertainty. In contrast, standard probabilistic models only predict certain anomaly probability, even when facing
> unseen inference data. Therefore, evidential learning facilitates a reliable generalization from a single frame to the complete abnormal interval through uncertainty estimation, which probabilistic models fail to do.
>
> **How Evidential Learning Works**: Evidential learning models the **distribution of probabilities** by assigning evidence to support predictions, rather than directly estimating probabilities. In our SF-VAD framework, we specifically apply this by fitting a **Beta distribution** to model anomaly evidence. The parameters of this Beta distribution directly quantify the strength of our belief in each prediction. Critically, this approach allows us to not only derive a point estimate of anomaly probability but also **explicitly quantify the uncertainty** associated with that prediction.
>
> **Evidential Learning's Advantages**: For epistemic uncertainty [5] estimation, previous methods [6,  7] adopt Bayesian Neural Network or Monte Carlo Dropout. Yet, these methods require multiple inferences to deduce the uncertainty and incur excessive computational cost. In contrast, evidential learning estimates uncertainty based on the Theory of Evidence and predicts both probability and uncertainty with a single forward pass, ensuring computational efficiency.
>
> ---
> **Response to Q5: Event-wise Robustness Test**
>
> For event-wise robustness test, we provide a performance comparison for video clips containing different numbers of disjoint abnormal events in XD-Violence.
> |AP/No. of Abnormal Events|1|2|3|4|5|
> |-|-|-|-|-|-|
> |Baseline|85.03|80.93|63.78|84.50|80.01|
> |FPL (Ours)|96.49|79.03|74.95|85.69|87.41|
>
> Regarding the request for UCF-Crime, we unfortunately find that the test set of UCF-Crime contains at most two disjoint abnormal events per abnormal video. Therefore, we are unable to assess AUC scores for video clips with three or more disjoint abnormal events as specifically requested by the reviewer due to this inherent dataset limitation.
> Still, to further demonstrate our method's robustness to varying numbers of abnormal events, we conduct additional experiments on UCF-Crime with the available event counts.
> |AUC/No. of Abnormal Events|1|2|
> |-|-|-|
> |Baseline|66.27|62.64|
> |FPL|73.86|80.73|
>
> FPL consistently outperforms baseline models under weak supervision across diverse numbers of abnormal events. It further verified the effectiveness of FPL in generalizing single-frame supervision into diverse abnormal events.
>
> ---
> **Response To Q6: Robustness to Non-ideal Annotations**
>
> To evaluate FPL's effectiveness under low-quality or inconsistent single-frame annotation, we further evaluate the performance of FPL using annotations from GlanceVAD [8] and ARG [9]. Fixing the hyperparameters, we randomly choose frames from glance annotation or a frame within the abnormal interval by the full annotations. The results are shown below.
> || SF Annotation | Glance$^*$ | Glance$^\dagger$ | Full$^*$ | Full$^\dagger$ |
> |-|-|-|-|-|-|
> |FPL|90.23|89.35|90.42|88.86|89.23|
>
> As shown in the table above, the performance of FPL remains consistent under different single frame settings, which demonstrates a good generalizability  under varying quality of annotations. The superior generalization performance comes from the dynamic abnormal event reasoning, where noise-free abnormal patterns are first modeled, then the precise abnormal intervals are mined based on the relevance and feature similarity to the learnt abnormal behaviors. In contrast to previous methods [10], which model the relative position within the abnormal interval by a fixed distribution, our method is much more **position-agnostic**, as abnormal intervals are dynamically deduced. This ensures excellent generalization capabilities, even in scenarios with ambiguous or non-representative single-frame annotations.
>
> ---
> [1] Micorek, Jakub, et al. "Mulde: Multiscale log-density estimation via denoising score matching for video anomaly detection." *CVPR*, 2024.
>
> [2] Ristea, Nicolae-C., et al. "Self-distilled masked auto-encoders are efficient video anomaly detectors." *CVPR*, 2024.
>
> [3] Yang, Guoqing, et al. "A multilevel guidance-exploration network and behavior-scene matching method for human behavior anomaly detection." *ACM MM*, 2024.
>
> [4] Shi, Haoyue, et al. "Learning Anomalies with Normality Prior for Unsupervised Video Anomaly Detection." *ECCV*, 2024.
>
> [5] Hüllermeier, Eyke, and Willem Waegeman. "Aleatoric and epistemic uncertainty in machine learning: An introduction to concepts and methods." *Machine learning*, 2021.
>
> [6] Subedar, Mahesh, et al. "Uncertainty-aware audiovisual activity recognition using deep bayesian variational inference." *ICCV*, 2019.
>
> [7] Zhang, Chen, et al. "Exploiting completeness and uncertainty of pseudo labels for weakly supervised video anomaly detection." *CVPR*, 2023.
>
> [8]  Zhang, Huaxin, et al. "Glancevad: Exploring glance supervision for label-efficient video anomaly detection." *arXiv*, 2024.
>
> [9] Liu, Kun, and Huadong Ma. "Exploring background-bias for anomaly detection in surveillance videos." *ACM MM*, 2019.
>
> [10] Cui, Ran, et al. "Video moment retrieval from text queries via single frame annotation." _SIGIR_, 2022.

---

> > ### Comment · Area_Chair_UFE4 · 2025-08-08
> >
> > Dear Reviewer mbCU,
> >
> > Could you please review the authors’ rebuttal and let them know if you have any further comments or concerns? Do you feel your original comments have been adequately addressed? If not, it would be helpful to highlight any remaining issues. I would be grateful if you could kindly engage in the discussion to help move the review process forward.
> >
> > Best regards,
> > AC

---

> > ### Comment · Reviewer_mbCU · 2025-08-08
> >
> > Thank you for your reply. I don't have any further questions.

---

> > > ### Author Response · Authors · 2025-08-09
> > > **Acknowledgement**
> > >
> > > We are truly delighted that the reviewer recognizes the novelty of our SF-VAD paradigm, the effectiveness of the proposed FPL framework, and the contribution of the SF-VAD benchmarks. We also appreciate the constructive feedback, which guides us in further validating our approach across different dimensions and strengthening the arguments presented in the paper.

---

### Official Review · Reviewer_2TVV · 2025-07-03

**Clarity:** 3
**Significance:** 2
**Originality:** 2
**Rating:** 4
**Confidence:** 3

**Summary:**

This paper proposes a new paradigm for video anomaly detection under single-frame supervision (SF-VAD), where only one anomalous frame per anomalous video needs to be manually labeled. The authors design a Frame-guided Progressive Learning (FPL) framework to generalize sparse single-frame supervision to event-level anomaly detection. Extensive experiments on multiple mainstream datasets demonstrate the method’s excellent trade-off between detection performance and labeling efficiency.

**Questions:**

Please respond to each item of the weaknesses mentioned above.

**Ethical Concerns:**

["NO or VERY MINOR ethics concerns only"]

**Final Justification:**

After reading the responses from the authors, I find that the supplemental results unfortunately reinforce my worries: with single-frame annotation, the model's core mechanism devolves into retrieving video segments most similar to the labeled anomalous frame. This reduction of VAD to a frame retrieval problem contradicts the field's core aim of achieving generalization and inductive reasoning across diverse contexts. This inherent limitation compromises the method's practical value for real-world scenarios involving unknown or complex anomalies.
After multiple rounds of dialogue, some of my concers have been addressed. Although there are still some viewpoints that I cannot fully agree with, I basically acknowledge the contribution of this method. Therefore, I decide to raise my rating score with a lower confidence.

**Limitations:**

Yes

**Paper Formatting Concerns:**

This paper is generally written well and I do not find obvious formatting issues in this paper.

**Quality:**

2

**Strengths And Weaknesses:**

Strengths:

1. The two-stage FPL framework combines evidence learning and feature similarity, expanding sparse labels to full anomaly event intervals.
2. The paper provides detailed performance comparisons and ablation studies, validating the method’s effectiveness.

Weaknesses & Questions:

1. The “single-frame annotation” setting actually contradicts the fundamental reality in VAD that anomalies are hard to annotate or acquire. In this setting, model learning shifts from “finding classification boundaries between normal and anomalous samples in many clips” to “finding the top-K segments most similar to a labeled anomalous frame within a single video”. This essentially transforms the anomaly detection problem into a retrieval task under strong supervision, weakening its generalization and induction challenge.
2. The choice of anomalous frame has a huge impact on performance. Annotators may not always select the most representative frames; labeling near event boundaries or ambiguous frames could hinder the model from learning typical anomaly patterns. The authors should detail the annotation process and provide case studies showing how the method handles boundary or unclear frames, and analyze the model’s robustness.
3. Some videos (e.g., in XD-Violence) may contain multiple types or stages of anomalies. Relying on a single frame is insufficient to cover the full semantic and temporal diversity of anomaly events. The authors should explain how their method handles multiple anomaly cases in detail.
4. I have concerns about the generalization ability of the proposed method. Compared to semi-supervised VAD, weakly-supervised VAD sacrifices labeling cost for greater generalization, yet this method’s training heavily relies on known anomalous frame features. If a novel anomaly type appears, or if the anomaly differs significantly from the labeled frame, detection performance may drop sharply—generalization is questionable.
5. The framework is based on the classic two-stage MIL for weakly-supervised VAD, but the model structure is overly complex and engineering-oriented, with limited originality.

---

> ### Author Rebuttal · Authors · 2025-07-31
>
> We sincerely appreciate the reviewer's meticulous feedback and profound insights into our method. Meanwhile, we are also grateful for the recognition of **SF-VAD's effectiveness, experimental completeness, and clarity of writing**. To address the concerns, we'd like to offer the following clarifications：
>
> ---
> **Response to Q1: Justification of SF-VAD**
>
> **Advantages of SF-VAD:**  SF-VAD significantly optimizes the utility of limited anomalous samples. On the one hand, it is **highly annotation efficient and accurate** . Since annotators must already perceive an obvious abnormal frame to label an entire video as abnormal. The additional annotation time is almost negligible:
> | Annotation Time| ShanghaiTech | UCF-Crime  | XD-Violence |
> | - | - | -| - |
> |Semi-supervised| 1.72 h| 87.66 h| 127.97 h|
> |Weakly-supervised|< 1.83 h|< 104.19 h |< 138.39 h|
> |Single-Frame supervised | 1.83 h|104.19 h| 138.39 h|
> |Fully-supervised|3.53h|127.54 h|217 h|
>
> Simultaneously, compared to fully-supervised methods, it avoids the inherent ambiguity of anomaly boundaries, resulting in remarkable annotation precision.
>
> On the other hand, SF-VAD delivers **precise anomaly reference**, which is absent in the weakly-supervised paradigm, facilitating **evidential and comprehensive anomaly learning** and resulting in **promising detection performance**:
> ||Supervision|ShanghaiTech|UCF-Crime|XD-Violence|
> |-|-|-|-|-|
> |ARG|Fully-supervised|-|82.0|-|
> |Ours|Fully-supervised|-|85.52|-|
> |AED-MAE|Semi-supervised|79.1|-|-|
> |MULDE|Semi-supervised|81.3|78.50|-|
> |CoMo|Weakly-supervised|97.60|86.10|81.30|
> |VadCLIP|Weakly-supervised|-|88.02|84.51|
> |**FPL (Ours)**|**Single-Frame supervised**|**98.32**|**90.23**|**89.56**|
>
> **Learning paradigm of FPL:** Leveraging single frame annotation, FPL effectively mines evidential abnormal intervals via integration of dynamic anomaly relevance reasoning and static feature similarity, **abandoning the groundless and notorious top-K selection** in MIL. As a result, a much clearer abnormal learning target is revealed, facilitating a more discriminative classification boundary.
>
> ---
> **Response to Q2: Model's Robustness to Shifting Annotations**
>
> **Authentic Annotation Protocol**: In the proposed three benchmarks, the annotators are allowed to freely navigate the video timeline and locate a random abnormal frame. Such annotation protocol ensures that the annotation **reflects genuine human interests** in realistic settings **without any prior guidance** that could artificially inflate performance. Details can be found in Sec. B in Supp.
>
> **Robustness to single-frame annotation bias**: To further validate the robustness of our method under varying single-frame annotation, we conduct additional experiments by sampling two set of abnormal frames from GlanceVAD [1] and ARG [2] in UCF-Crime:
> ||SF Annotation|Glance$^*$ |Glance$^\dagger$|Full$^*$ |Full$^\dagger$|
> |-|-|-|-|-|-|
> |FPL|90.23|89.35|90.42|88.86|89.23|
>
> The performance of FPL remains mostly consistent with randomly selected ambiguous or non-representative single-frame annotations. In contrast to previous methods [1, 3] with a fixed distribution, our method is much more **position-agnostic**, as abnormal intervals are dynamically deduced.
>
> ---
> **Response Q3: Effectiveness in Multi-event Scenarios**
>
> We appreciate the reviewer for highlighting multi-event issues, which are
> pivotal for realizing robust anomaly detection. Specifically, we tailor an abnormal event mining algorithm for multi-interval anomaly mining, which integrates anomaly relevance and feature similarity. This algorithm first selects potential abnormal frames by assessing the similarity across both action and appearance features, with the selection further refined by variance verification and relative gap constraints to enhance exploration accuracy. It then utilizes these pending frames to gather similar frames into the intervals by estimating anomaly relevance.
> For validation, we present ablation experiments on multi-event anomalies in XD-Violence:
> |AP/No. of Abnormal Events|1|2|3|4|5|
> |-|-|-|-|-|-|
> |Baseline|85.03|80.93|63.78|84.50|80.01|
> |FPL (Ours)|96.49|79.03|74.95|85.69|87.41|
>
> For further evaluation, we extend our evaluation to include multi-event experiments on UCF-Crime. The results are as follows:
> |AUC/No. of Abnormal Events|1|2|
> |-|-|-|
> |Baseline|66.27|62.64|
> |FPL (Ours)|73.86|80.73|
>
> These experimental results further substantiate our method's generalizability in capturing patterns from multiple abnormal segments, which is primarily attributed to multi-event anomaly mining algorithm.
>
> ---
> **Response to Q4: Investigation on Generalization**
>
> For a comprehensive generalization evaluation, we consider the following six aspects, ordered by importance:
>
> **Performance Consistency across Various Domains**: it directly measures the model's generalizability across different real-world distributions. As shown in Tab. 1 and Tab. 2, our method **consistently outperforms SOTA methods** with higher AUC and lower False Alarm Rate across a wide spectrum of domains, anomaly behaviors, perspectives, and data sources.
>
> **Detection of Diverse Anomaly Types**: we provide a class-wise AUC analysis on UCF-Crime:
> ||Arrest|Arson|Assault|Burglary|Vandalism|Explosion|Shooting|
> |-|-|-|-|-|-|-|-|
> |UMIL|68.7|64.9|70.3|72.1|77.4|65.0|-|
> |RTFM|66.1|61.2|69.3|**72.9**|63.3|49.5|71.7|
> |CoMo|56.2|47.5|74.1|69.4|83.8|64.6|75.1|
> |**FPL (Ours)** |**70.8**|**71.7**|**81.5**|72.3|**85.8**|**70.3**|**82.9**|
>
> In addition, we further extend our evaluation to XD-Violence:
> ||Abuse|Explosion|Riot|Fighting|overall|
> |-|-|-|-|-|-|
> |PEL4VAD|70.5|63.9|96.5|**83.8**|85.26|
> |**FPL (Ours)** |**79.41**|**68.22**|**97.63**|81.33|**89.56**|
>
> These results further corroborate our method's robust generalization across diverse abnormal behaviors, including both prominent categories like *Riot* and visual concealed behaviors such as *Abuse*.
>
> **Effectiveness in Multi-event Scenarios**: Please refer to Q3.
>
> **Robustness to Single-frame Annotation Bias**: Please refer to Q2.
>
> **Adaptability across Different Backbones**: To further evaluate FPL's effectiveness with varied backbone architectures, we conduct an additional experiment utilizing a CNN-based backbone from CU-Net [4] in UCF-Crime. The results are presented below:
> ||Supervision|Weak Label|SF Annotation|Glance|Full|Parameters|
> |-|-|-|-|-|-|-|
> |VadCLIP|Weakly-supervised (with text features)|88.02|-|-|-|162.26 M|
> |CU-Net (with FPL)|Single-Frame supervised|86.22|89.35|88.76|87.30|0.80 M|
> |FPL (Ours)|Single-Frame supervised|83.67|90.23|90.42|89.23|2.21 M|
>
> As the results indicate, FPL surpasses SOTA methods with both CNN-based and Transformer-based lightweight backbones, demonstrating ideal adaptability across different model architectures.
>
> **Cross-dataset Transferability**: Due to the inconsistency of anomaly definitions, cross-dataset validation resembles an **open-set problem**, rather than evaluation of generalization. Nevertheless, to thoroughly address the reviewer's request, we conduct additional experiments:
> |Source|UCF-Crime|XD-Violence|UCF-Crime|XD-Violence|ShanghaiTech|UCF-Crime|XD-Violence|
> |-|-|-|-|-|-|-|-|
> |Target|→|UCF-Crime|XD-Violence|←|→|ShanghaiTech|←|
> |RTFM|84.48|68.59 (18.81% ↓)|76.62|37.30 (51.32% ↓)|97.20|45.49 ( 53.2% ↓)|-|
> |CoMo|86.07|69.89 ( 18.80% ↓)|81.31|46.74 ( 42.52% ↓)|97.59|52.02 ( 46.7% ↓)|-|
> |FPL (Ours)|90.23|85.67 (5.05% ↓)|89.56|82.31 (8.10% ↓)|98.32|94.21 (4.18% ↓)|-|
>
> The results indicate a commendable cross-dataset transferability. This is primarily because single-frame supervision provides a clean and precise learning target. Instead, the learning process of previous MIL-based methods is vulnerable to shifted domains.
>
> ---
> **Response to Q5: Difference from Previous 2-stage MIL Methods**
>
> Previously, weakly-supervised VAD methods [4, 5] employed a 2-stage training strategy with pseudo labels, attempting to alleviate impact of noise. However, such methods inevitably converge to erroneous anomaly patterns with groundless top-k selection due to the deficiency of coarse-grained weak labels.
> In contrast, FPL first introduces evidential learning to estimate the anomaly relevance to the annotated precise abnormal patterns. Subsequently, intact abnormal intervals are mined based on the dynamic anomaly relevance and static feature similarity to reveal an accurate abnormal learning target. This resolves the issue of indistinct classification boundary caused by top-k selection, resulting in higher AUC and lower false alarms.
>
> ---
> **Response to Q6: Lightweight Architecture Design**
> In fact, our model is concise with effective temporal modeling as in Sec. F in Supp. The model parameter size (in million) and inference time on the UCF-Crime test set are listed as follows:
> |Methods| Supervision| Parameters (↓) | Inference Time (↓) | UCF-Crime AUC |
> |-|-|-|-|-|
> |Conv-AE|Semi-supervised|10.14 M|-|-|
> |MGFN| Weakly-supervised|28.46 M|3.70 s|86.67|
> |RTFM| Weakly-supervised|24.72 M|3.23 s|84.30|
> |UR-DMU|Weakly-supervised| 6.49 M|12.39 s|86.97|
> |VadCLIP|Weakly-supervised (with Text Features) |162.26 M|5.65 s|88.02
> | **FPL(Ours)** | Single-Frame supervised|**2.21 M**|**0.51 s**|**90.23**|
>
> Remarkably, FPL  enables a lightweight network with a **smaller parameter size** and much **faster inference time**, to capture **more accurate anomaly cues**.
>
> ---
> [1] Zhang, Huaxin, et al. "Glancevad: Exploring glance supervision for label-efficient video anomaly detection." *arXiv*, 2024.
>
> [2] Liu, Kun, and Huadong Ma. "Exploring background-bias for anomaly detection in surveillance videos." *ACM MM*, 2019.
>
> [3] Cui, Ran, et al. "Video moment retrieval from text queries via single frame annotation." *SIGIR*, 2022.
>
> [4] Zhang, Chen, et al. "Exploiting completeness and uncertainty of pseudo labels for weakly supervised video anomaly detection." *CVPR*, 2023.
>
> [5] Li, Shuo, et al. "Self-training multi-sequence learning with transformer for weakly supervised video anomaly detection." *AAAI*, 2022.

---

> ### Comment · Reviewer_2TVV · 2025-08-08
>
> Thanks to the authors for the additional experiments and detailed explanations addressing my concerns. Unfortunately, the supplemental experimental results seem to further confirm my worries: under the single-frame annotation setting, the core of the model essentially becomes retrieving segments most similar to the annotated anomalous frame within a video, which fundamentally reduces the VAD task to a frame retrieval problem. This approach contradicts the fundamental goal of video anomaly detection, which is to achieve generalization and inductive reasoning across diverse scenarios, and thus diminishes the method’s practical value in real-world settings involving unknown or complex anomalies.

---

> > ### Author Response · Authors · 2025-08-08
> > **Further Response on the Learning Paradigm and Generalization**
> >
> > We appreciate the reviewer's response and the opportunity to clarify our work. We respectfully disagree with the assessment of SF-VAD. As reviewers BiMk,  ZVwY, and mbCU state, SF-VAD represents a novel and effective VAD paradigm that provides a principled solution to a long-standing challenge in weakly-supervised learning. The key distinction lies in the introduction of a **grounded abnormal behavior reference**, which **enables more robust anomaly learning by exploiting anomaly relevance and feature similarity, instead of groundless top-k sampling**. Experimental results have since verified our SF-VAD method's superior detection performance and strong generalization across multiple anomaly domains, categories, and network architectures, providing a feasible solution for real-world applications.
> >
> > The further response is structured into three key aspects: theoretical justification, clarifying the paradigm's objective, and experimental validation.
> >
> > ---
> > **Theoretical Justification: A Grounded Learning Paradigm**
> >
> > | VAD Paradigm               | Abnormal Behavior Reference | Anomaly Learning Source                                 |
> > | --------------------------- | --------------------------- | ------------------------------------------------------- |
> > | Semi-supervised VAD         | ✕                           | -                                                       |
> > | Weakly-supervised VAD       | ✕                           | Groundless Top-k Clips                                  |
> > | Single-Frame supervised VAD | ✓                           | Clips Mined by Anomaly Relevance and Feature Similarity |
> >
> > **The Challenge of Weakly-supervised VAD Paradigm**
> >
> > The long-standing challenge for well-established weakly-supervised VAD has been to learn anomalies from more reliable video clips, but this is difficult without a clear abnormal behavior reference. Therefore, existing methods have to rely on **groundless top-k sampling** based on randomly initialized anomaly scores. This makes the models susceptible to learning imprecise anomaly patterns from noisy or interfering clips.
> > It further limits robustness and generalizability, since these models tend to omit subtle anomalies and struggle with background interference.
> >
> > **The SF-VAD Solution**
> >
> > Instead, the proposed SF-VAD paradigm directly solves this problem by providing a **grounded abnormal behavior reference**. Leveraging this precise reference, our method learns the decision boundary from clips that are reliably mined by dynamic anomaly relevance and static feature similarity, directly addressing the limitations of groundless sampling. This principled approach ensures a much more accurate and reliable anomaly learning target.
> >
> > ---
> > **Rebuttal for the Reviewer's Assumption of Frame Retrieval**
> >
> > We respectfully disagree with the assumption that our method is a frame retrieval task. Rather, our objective is to **generalize the supervision from a single annotated frame to the entire abnormal interval** through progressive learning of dynamic anomaly relevance and static feature similarity. As a result, it facilitates robust anomaly learning from more credible abnormal intervals, moving beyond frame retrieval.
> >
> > SF-VAD does not weaken the inductive challenge inherent to video anomaly detection. The core difficulty in weakly-supervised VAD lies in learning from credible abnormal clips.
> > Prior works have often relied on workaround strategies, such as multi-stage training, to suppress noise. Unfortunately, in the absence of **abnormal behavior references**, it is nearly impossible to accurately deduce anomalous frames. Thus, these approaches inevitably depend on **ungrounded top-k sampling**, leading to noisy and imprecise decision boundary. It is especially problematic for subtle anomalies and in scenarios with severe background distractions.
> >
> > In contrast, SF-VAD introduces explicit **abnormal behavior references**, enabling the model to perform **more meaningful and evidence-based inductive reasoning**. Instead of reducing the task to simple frame retrieval, this design supports the discovery of reliable abnormal patterns and enhances generalization to unknown or complex anomaly scenarios.

---

> > ### Author Response · Authors · 2025-08-08
> > **Experimental Validation of Generalizability in Practical Settings**
> >
> > To comprehensively evaluate the generalization ability of our method, we design rigorous experiments from six perspectives:
> >
> > 1. **Performance across different domains**: Our method consistently surpasses prior SOTA approaches on the challenging ShanghaiTech, UCF-Crime, and XD-Violence, demonstrating superior performance under diverse perspectives and anomaly distributions, highlighting FPL’s robustness across real-world variations.
> > ||Supervision|ShanghaiTech|UCF-Crime|XD-Violence|
> > |-|-|-|-|-|
> > |ARG|Fully-supervised|-|82.0|-|
> > |Ours|Fully-supervised|-|85.52|-|
> > |AED-MAE|Semi-supervised|79.1|-|-|
> > |MULDE|Semi-supervised|81.3|78.50|-|
> > |CoMo|Weakly-supervised|97.60|86.10|81.30|
> > |VadCLIP|Weakly-supervised|-|88.02|84.51|
> > |**FPL (Ours)**|**Single-Frame supervised**|**98.32**|**90.23**|**89.56**|
> >
> > 2. **Detection of diverse anomaly categories**: We perform class-wise analysis on UCF-Crime and XD-Violence, covering both visually obvious events and subtle ones. Results show FPL achieves the best or highly competitive AUCs across all categories, verifying its capability to model different types of anomalous behaviors.
> > ||Arrest|Arson|Assault|Burglary|Vandalism|Explosion|Shooting|
> > |-|-|-|-|-|-|-|-|
> > |UMIL|68.7|64.9|70.3|72.1|77.4|65.0|-|
> > |RTFM|66.1|61.2|69.3|72.9|63.3|49.5|71.7|
> > |CoMo|56.2|47.5|74.1|69.4|83.8|64.6|75.1|
> > |FPL (Ours) |70.8|71.7|81.5|72.3|85.8|70.3|82.9|
> >
> > 3. **Multi-event anomaly detection**: Extensive experiments on videos containing multiple distinct anomaly events confirm FPL’s ability to locate and model multiple abnormal intervals effectively, without relying on ground-truth segmentation. This is enabled by our adaptive interval mining strategy.
> > |AP/No. of Abnormal Events|1|2|3|4|5|
> > |-|-|-|-|-|-|
> > |Baseline|85.03|80.93|63.78|84.50|80.01|
> > |FPL (Ours)|96.49|79.03|74.95|85.69|87.41|
> >
> > 4. **Robustness to single-frame annotation bias**: To simulate potential human bias during annotation, we evaluate performance under various single-frame selections. FPL maintains stable accuracy, showing its strong tolerance to annotation shift and confirming the reliability of our dynamic relevance-based mining.
> > ||SF Annotation|Glance$^*$ |Glance$^\dagger$|Full$^*$ |Full$^\dagger$|
> > |-|-|-|-|-|-|
> > |FPL|90.23|89.35|90.42|88.86|89.23|
> >
> > 5. **Adaptability to different backbones**: We further test FPL on both Transformer-based and CNN-based architectures. Despite a significant reduction in model size and complexity, FPL still delivers competitive or superior performance, showing its adaptability and scalability to diverse deployment settings.
> > ||Supervision|Weak Label|SF Annotation|Glance|Full|Parameters|
> > |-|-|-|-|-|-|-|
> > |VadCLIP|Weakly-supervised (with text features)|88.02|-|-|-|162.26 M|
> > |CU-Net (with FPL)|Single-Frame supervised|86.22|89.35|88.76|87.30|0.80 M|
> > |FPL (Ours)|Single-Frame supervised|83.67|90.23|90.42|89.23|2.21 M|
> >
> >
> > 6. **Cross-dataset transferability**: Cross-domain experiments reveal that FPL suffers minimal performance drop compared to large degradations in previous methods. This suggests that our FPL approach facilitates stronger generalization, even under open-set domain shifts.
> > |Source|UCF-Crime|XD-Violence|UCF-Crime|XD-Violence|ShanghaiTech|UCF-Crime|XD-Violence|
> > |-|-|-|-|-|-|-|-|
> > |Target|→|UCF-Crime|XD-Violence|←|→|ShanghaiTech|←|
> > |RTFM|84.48|68.59 (18.81% ↓)|76.62|37.30 (51.32% ↓)|97.20|45.49 ( 53.2% ↓)|-|
> > |CoMo|86.07|69.89 ( 18.80% ↓)|81.31|46.74 ( 42.52% ↓)|97.59|52.02 ( 46.7% ↓)|-|
> > |FPL (Ours)|90.23|85.67 (5.05% ↓)|89.56|82.31 (8.10% ↓)|98.32|94.21 (4.18% ↓)|-|
> >
> > In summary, the breadth and consistency of results across these six aspects affirm the strong generalization capability of SF-VAD under realistic, diverse, and challenging settings. The results manifest that SF-VAD offers a more reliable and practical solution for real-world video anomaly detection applications.

---

> > > ### Comment · Area_Chair_UFE4 · 2025-08-08
> > >
> > > Dear Reviewer 2TVV
> > > Could you please review the authors’ rebuttal and let them know if you have any further comments or concerns? Do you feel your original comments have been adequately addressed? If not, it would be helpful to highlight any remaining issues.
> > >
> > > Best
> > > AC

---

> > > ### Comment · Reviewer_2TVV · 2025-08-09
> > >
> > > Could you please explain why the performance with Single-Frame supervision can be much better than fully-supervision?

---

> > > > ### Author Response · Authors · 2025-08-09
> > > > **Comparison with Fully-supervised Paradigm**
> > > >
> > > > The reviewer gains a nuanced insight and points out a rather interesting and seemingly counter-intuitive topic that the proposed SF-VAD outperforms fully-supervised paradigm significantly. The detailed explanations are organized in the following two sections: *weakness of fully-supervised VAD paradigm*, *advantages of SF-VAD*.
> > > >
> > > > ---
> > > > **Weaknesses of Fully-supervised VAD Paradigm**
> > > >
> > > > In VAD field, fully-supervised paradigm doesn't perform ideally, which is also observed in other literature [1, 2]. The key reasons lie in following three aspects: annotation dilemma, generalization issue, and data imbalance.
> > > >
> > > > First, the annotation of precise abnormal intervals faces fundamental dilemmas, including **inconsistent boundary annotation, omission of multi-interval abnormal events**. On the one hand, the definition of abnormal event boundary is ambiguous, which leads to imprecise and inconsistent annotation, as mentioned in lines 28-31. On the other hand, an abnormal video may contain multiple abnormal events, and previous annotations [3, 4] often face the omission of anomalies due to the heavy burden of fine-grained annotation. This problem is particularly pronounced in fully supervised learning, where models tend to fit strictly to the provided labels. Consequently, **the integration of inferior annotation and strict constraint during training leads to ambiguous and inconsistent decision boundaries**, failing to learn coherent abnormal patterns.
> > > >
> > > > Second, the fully-supervised paradigm suffers from **severe generalizability issues**. Abnormal behaviors within the same domain can vary substantially, yet fully-supervised approaches enforce strict supervision during training. This often causes overfitting to the limited patterns seen in the training data. The problem is further exacerbated by the small scale of existing fully-supervised VAD datasets, constrained by the high cost of precise annotations. Dataset statistics are provided in **Sec.C of the Supp**.
> > > >
> > > > Third, VAD datasets face **severe data imbalance**. Since abnormal behaviors are inherently difficult to collect, the vast majority of frames in VAD datasets are normal. Given that fully-supervised models fit all labels, this imbalance causes them to exhibit **a strong bias towards predicting frames as normal**. Such bias can undermine the performance of objective evaluation metrics such as AUC, whose computation emphasizes the correct discrimination of abnormal frames, thereby leading to lower scores under severe imbalance.
> > > >
> > > > Due to the above issues, the experimental results and application prospects of the fully-supervised paradigm are unfortunately not ideal. Additional ablations are conducted:
> > > > |Methods|Supervision|UCF-Crime|
> > > > |-|-|-|
> > > > |ARG|Fully-supervised|82.0|
> > > > |**Baseline**|Fully-supervised|85.52|
> > > > |LANP|Semi-supervised|80.02|
> > > > |UR-DMU|Weakly-supervised|86.97|
> > > > |**FPL**|Single-frame supervised|90.23|
> > > >
> > > > These results further corroborate that the limitations discussed above substantially hinder the effectiveness of fully-supervised methods. **More comprehensive visual comparisons and ablation analyses will be incorporated in the revision**.
> > > >
> > > > ---
> > > > **Advantages of SF-VAD**
> > > >
> > > > In contrast, SF-VAD fundamentally circumvents these issues by precisely annotating only a single abnormal frame, thereby **reducing annotation cost**, **ensuring annotation accuracy**, **mitigating annotation bias**, **improving generalizability**, **alleviating data imbalance**.
> > > >
> > > > The single-frame annotation substantially decreases labeling effort compared to dense frame-level labeling, while avoiding errors from ambiguous temporal boundaries by precisely marking a definitive abnormal instance. This strategy inherently reduces subjective biases related to boundary definitions. Furthermore, the proposed Frame Propagation Learning (FPL) enables leveraging a single annotated frame to infer multiple abnormal segments, minimizing manual labeling demands.
> > > >
> > > > Training under a framework similar to weak supervision, SF-VAD focuses on learning discriminative abnormal features without overfitting to exhaustive annotations, thus improving generalizability across anomaly types and domains. Additionally, its loss function balances contributions from normal and abnormal samples, counteracting the bias toward normal predictions caused by severe data imbalance.
> > > >
> > > > Overall, SF-VAD demonstrates marked advantages in annotation efficiency, detection accuracy, robustness, and practical applicability.
> > > >
> > > > ---
> > > > [1] Samaila, Yau Alhaji, et al. "Video anomaly detection: A systematic review of issues and prospects." _Neurocomputing_, 2024.
> > > >
> > > > [2] Mostafa, Iman, et al. "Abnormal Human Activity Recognition in Video Surveillance: A Survey." _Port-Said Engineering Research Journal_, 2024.
> > > >
> > > > [3] Liu, Kun, and Huadong Ma. "Exploring background-bias for anomaly detection in surveillance videos." _ACM MM_, 2019.
> > > >
> > > > [4] Landi, Federico, Cees GM Snoek, and Rita Cucchiara. "Anomaly locality in video surveillance." _arXiv_, 2019.

---

### Official Review · Reviewer_ZVwY · 2025-07-03

**Clarity:** 3
**Significance:** 2
**Originality:** 2
**Rating:** 4
**Confidence:** 3

**Summary:**

The authors introduce a new annotation regime for video-anomaly detection, Single-Frame supervised VAD, in which each abnormal video is annotated with only one abnormal frame.  They (i) re-label three popular benchmarks—ShanghaiTech, UCF-Crime and XD-Violence—to create SF-VAD testbeds, and propose Frame-guided Progressive Learning.

**Questions:**

If you train on ShanghaiTech SF-VAD but test on UCF-Crime without fine-tuning, how does performance change?  This relates to my previous question on the single-frame anomaly generalization.

Does FPL still help when using a lightweight CNN or ViT backbone?

**Ethical Concerns:**

["NO or VERY MINOR ethics concerns only"]

**Final Justification:**

I thank authors' response. The rebuttal addressed most of my concerns, and I still see the novelty of this approach and new setup. I will remain my rating for the paper.

**Limitations:**

Yes

**Quality:**

3

**Strengths And Weaknesses:**

This paper formalise a new anomaly detection benchmark: Single-Frame supervised VAD. Which allevaite siginificant amount of annotation works but achieve good performances.

Weakness:

The mining seem to relies on several thresholds. Could the authors provide some insight or ablations to how to choose the best thresholds.

The performance of this Single-Frame-level supervision achieve higher performances than the fully supervised approaches. This seems counter intuitive. Could the authors leverage on this and discuss the potential reasons?

One of the argument for anomaly detection is that the normal events are more controlled and common in real world. Hence, annotating the normal video/frames is a lot easier than abnormals. Collecting the annotation for single frame anomalies are also time-consuming from my perspective, you need to locate which frame from a raw video contain anomalies, which some times even harder than the video annotations in the WVAD setups. This could be alleviate if the model trained on certain type of single frame anomalies could generalize to unseen anomalous events.

---

> ### Author Rebuttal · Authors · 2025-07-31
>
> We sincerely appreciate the reviewer's thorough and insightful assessment, particularly the recognition of the **effectiveness of SF-VAD**. To respond the constructive questions, we'd like to offer the following clarifications：
>
> ---
> **Response to Q1: Choice of Thresholds**
>
> In *Abnormal Event Mining*, we utilize thresholds $\mathbf{\theta}$ to control the key frame selection process by frame-wise feature similarity.
> First, $\theta_1$ serves as a critical threshold to evaluate the discriminative capability of the feature similarity for identifying distinct abnormal intervals. A low variance suggests a consistent scene and motion pattern throughout the video, implying a reduced likelihood of multiple distinct abnormal intervals and thus limiting the reliability of multi-interval probing. In such instances, the system adaptively shifts its focus to precisely mine similar frames in the vicinity of the single annotated abnormal frame. The hyperparameter analysis of $\theta_1$​ is shown below:
> |$\theta_1$|0.05|0.1|0.15|0.2|
> |-|-|-|-|-|
> |UCF|88.62|90.23|89.38|86.23|
> |XD|87.11|89.56|88.52|86.27|
>
> Second, $\theta_2$ controls the degree of similarity required. A greater value of  $\theta_2$ indicates that only frames exhibiting a higher degree of similarity are considered as alike abnormal clips.
> |$\theta_2$|0.7|0.8|0.9|0.95|0.98|
> |-|-|-|-|-|-|
> |UCF|87.56|88.12|89.38|90.23|88.86|
> |XD|87.96|88.25|88.43|89.56|89.04|
>
> Third, $\theta_3$ controls the gap of the relative position of selected key frames, which delegates the prominent key frames within each abnormal interval. Considering both computational overhead and mining precision, the key frame selection process is designed to sample only the most crucial frame within each abnormal interval for subsequent interval mining. Consequently, the gap between selected key frames is managed.
> |$\theta_3$|0.1|0.2|0.3|0.4|0.5|
> |-|-|-|-|-|-|
> |UCF|89.46|90.23|89.18|89.33|88.12|
> |XD|89.02|89.56|88.92|88.78|87.84|
>
> As the results show, the setting of thresholds $\theta$ achieve an ideal trade-off between interval mining completeness and accuracy, and a preferred generalization capability over multi-interval anomalies as Fig. 5 shows.
>
> ---
> **Response to Q2: Comparison with Fully-supervised Paradigm**
>
> The reviewer gains a nuanced insight and points out a rather interesting topic in VAD fields that fully-supervised paradigm doesn't perform ideally. Noticing such results are also observed in other literature [1, 2]. The key reasons lie in following aspects.
>
> First, the annotation of precise abnormal intervals faces fundamental dilemmas, **including inconsistent boundary annotation, omission of multi-interval abnormal events**. In the one hand, the definition of abnormal event boundary is ambiguous. For instance, the interweaving of flame and smoke preceding and after the *explosion* anomaly are hard to categorize. The ambiguity leads to imprecise and inconsistent annotation, as mentioned in lines 28-31. It hinders the learning of a coherent decision boundary. On the other hand, an abnormal video may contain multiple abnormal events, and previous annotations [3, 4] often face the omission of anomalies due to the heavy burden of fine-grained annotation. This problem is particularly pronounced in fully supervised learning, where models tend to fit strictly to the provided labels. Consequently, some anomalies might be incorrectly classified as normal, leading to severe convergence issues during training.
>
> Second, the fully-supervised paradigm faces **severe generalizability issues**. As abnormal behaviors can vary greatly within a domain (e.g., *Explosion*,  *Riot* in UCF-Crime), fully-supervised approaches impose strict supervision during training, which often leads to problems like overfitting. Moreover, the limited scale of fully-supervised VAD datasets due to high annotation costs exacerbates this issue. Details of the dataset statistics can be found in **Sec. C in Supp**.
>
> Third, VAD datasets face **severe data imbalance**. The duration of abnormal videos and normal videos in the train set is shown below:
> |Benchmark |Duration of anomaly|Duration of normalcy |
> |-|-|-|
> |ShanghaiTech|0.26 h|1.72 h|
> |UCF-Crime|29.40 h|87.66 h|
> |XD-Violence|61.27 h|127.97 h|
>
> Since abnormal behaviors are inherently difficult to collect, the vast majority of frames in VAD datasets are normal. Given that fully-supervised models fit all labels, this imbalance causes them to exhibit a strong bias towards predicting frames as normal.
>
> Due to the above issues, the experimental results and application prospects of the fully-supervised paradigm are unfortunately not ideal. Additional ablations are conducted:
> |Methods|Supervision|UCF-Crime|
> |-|-|-|
> |ARG|Fully-supervised|82.0|
> |**Baseline**|Fully-supervised|85.52|
> |LANP|Semi-supervised|80.02|
> |UR-DMU|Weakly-supervised|86.97|
> |**FPL**|Single-frame supervised|90.23|
>
> In Contrast, SF-VAD fundamentally circumvents these issues by precisely annotating only a single abnormal frame, thereby **reducing annotation cost**, **ensuring accuracy**, and **mitigating annotation bias**. Through its exploration of anomaly pattern learning, it achieves  generalizability in multiple aspects and by inherently balancing normal and abnormal samples in its loss function, it demonstrates promising capabilities across annotation cost, detection performance, generalizability, and real-world applications.
>
> ---
> **Response to Q3: Annotation Efficiency**
>
> In fact, the addition of annotation cost of SF-VAD compared to weakly-supervised VAD is minor. As described in Sec.B in the Supp., the annotators are allowed to freely navigate in the video to locate a random abnormal frame. As people must witness an abnormal frame first, before they can make sure that the video is abnormal, the time addition is negligible. Dataset statistic in Sec. 4.2 is aligned with the protocol where the annotated abnormal frames is located at the front of the abnormal interval mostly.
>
> ---
> **Response to Q4: Method's Robustness**
>
> To address the reviewer's concerns, we evaluate the robustness of our method from the following aspects.
> **Robustness to Single-Frame Annotation Bias:** we conduct additional experiments to validate our method's robustness under varying single-frame annotation qualities. We sampled abnormal frames from GlanceVAD [6] and ARG [3] in UCF-Crime:
> ||SF Annotation|Glance$^*$ |Glance$^\dagger$|Full$^*$ |Full$^\dagger$|
> |-|-|-|-|-|-|
> |FPL|90.23|89.35|90.42|88.86|89.23|
>
> The results show that FPL's performance remains highly consistent, even with randomly selected ambiguous or non-representative single-frame annotations. Unlike previous methods [6, 7] that rely on a fixed distribution, my approach is significantly more **position-agnostic** because it dynamically deduces abnormal intervals. This design intrinsically enhances robustness against annotation variability.
>
> **Cross-Dataset Transferability:** While cross-dataset validation often presents an **open-set problem** due to inconsistent anomaly definitions, we perform additional experiments to rigorously address the reviewer's query regarding single-frame anomaly generalization:
> |Source|ShanghaiTech|UCF-Crime|XD-Violence|UCF-Crime|XD-Violence|ShanghaiTech|UCF-Crime|XD-Violence|
> |-|-|-|-|-|-|-|-|-|
> |Target|UCF-Crime|→|UCF-Crime|XD-Violence|←|→|ShanghaiTech|←|
> |RTFM|-|84.48|68.59 (18.81% ↓)|76.62|37.30 (51.32% ↓)|97.20|45.49 ( 53.2% ↓)|-|
> |CoMo|-|86.07|69.89 ( 18.80% ↓)|81.31|46.74 ( 42.52% ↓)|97.59|52.02 ( 46.7% ↓)|-|
> |FPL (Ours)|90.23 65.86 (27.01% ↓)|90.23|85.67 (5.05% ↓)|89.56|82.31 (8.10% ↓)|98.32|94.21 (4.18% ↓)|-|
>
> Given the fundamental difference in anomaly definitions between ShanghaiTech and UCF-Crime (where behaviors like biking, skateboarding, and chasing are considered abnormal in ShanghaiTech but normal in UCF-Crime), the first cross-dataset result shows a notable decrease. However, in all other cross-dataset evaluations, results **demonstrate a commendable cross-dataset transferability** for FPL. This superior generalization capability stems from the proposed single-frame supervision, which provides a clean and precise learning target. In contrast, the learning processes of previous MIL-based methods are often vulnerable to domain shifts, highlighting FPL's advantage in adapting to unseen anomalous events.
>
> ---
> **Response to Q5: Effectiveness with Different Backbones**
>
> To further evaluate the effectiveness of FPL using different backbones, we add additional experiments using CNN-base network from CU-Net [5], the results are shown below.
> ||Supervision|Weak Label|SF Annotation|Glance|Full|Parameters|
> |-|-|-|-|-|-|-|
> |VadCLIP|Weakly-supervised (with text features)|88.02|-|-|-|162.26 M|
> |CU-Net (with FPL)|Single-Frame supervised|86.22|**89.35**|88.76|87.30|0.80 M|
> |FPL (Ours)|Single-Frame supervised|83.67|**90.23**|90.42|89.23|2.21 M|
>
> As the result shows, FPL surpasses SOTA methods with a lightweight CNN backbone. The superiority of performance mostly comes from the accurate learning target enabled by the FPL, instead of complex backbone design.
>
> [1] Samaila, Yau Alhaji, et al. "Video anomaly detection: A systematic review of issues and prospects." *Neurocomputing*, 2024.
>
> [2] Mostafa, Iman, et al. "Abnormal Human Activity Recognition in Video Surveillance: A Survey." *Port-Said Engineering Research Journal*, 2024.
>
> [3] Liu, Kun, and Huadong Ma. "Exploring background-bias for anomaly detection in surveillance videos." *ACM MM*, 2019.
>
> [4] Landi, Federico, Cees GM Snoek, and Rita Cucchiara. "Anomaly locality in video surveillance." *arXiv*, 2019.
>
> [5] Zhang, Chen, et al. "Exploiting completeness and uncertainty of pseudo labels for weakly supervised video anomaly detection." *CVPR*, 2023.
>
> [6] Zhang, Huaxin, et al. "Glancevad: Exploring glance supervision for label-efficient video anomaly detection." *arXiv*, 2024.
>
> [7] Cui, Ran, et al. "Video moment retrieval from text queries via single frame annotation." *SIGIR*, 2022.

---

> > ### Comment · Reviewer_ZVwY · 2025-08-07
> >
> > I thank authors' response. The rebuttal addressed most of my concerns, and I still see the novelty of this approach and new setup. I will remain my rating for the paper.

---

> > > ### Author Response · Authors · 2025-08-08
> > > **Acknowledgment**
> > >
> > > Thank you for the constructive and insightful feedback throughout the review process. We sincerely appreciate your recognition of the proposed SF-VAD paradigm.

---

### Official Review · Reviewer_BiMk · 2025-07-05

**Clarity:** 2
**Significance:** 3
**Originality:** 2
**Rating:** 4
**Confidence:** 3

**Summary:**

The paper introduces an innovative Single-Frame Supervised Video Anomaly Detection (SF-VAD) paradigm that significantly enhances annotation efficiency by utilizing just one annotated abnormal frame per abnormal video, substantially reducing the annotation workload compared to traditional fully-supervised methods. To generalize this sparse supervision for comprehensive understanding of entire abnormal events, the authors propose the Frame-Guided Progressive Learning (FPL) framework, which consists of two stages: first estimating the relevance of each frame to the annotated abnormal frame, and then extending this supervision to multiple abnormal events while decoupling normal patterns to minimize false alarms. The method is validated on three newly constructed SF-VAD benchmark datasets, demonstrating superior performance in detecting both obvious and subtle anomalies with reduced false alarm rates.

**Questions:**

Clarification on Dataset Configuration:
Description: The paper mentions constructing three SF-VAD benchmark datasets but does not provide detailed information on how these datasets were configured, including data sources, annotation process, and data splits.
Suggestion: Please provide a comprehensive description of the dataset configuration in the supplementary material or an appendix. Include details such as the criteria for selecting videos, the annotation process (e.g., how annotators were instructed to label the first abnormal frame), and any preprocessing steps applied to the data.
Impact on Evaluation: If this information is provided, it will significantly enhance the reproducibility of your work and allow other researchers to build upon your datasets. This could increase my evaluation score if the dataset configuration is well-documented and transparent.
Discussion on Method Limitations:
Description: While the paper highlights the advantages of the SF-VAD paradigm, it does not fully discuss its limitations or potential failure cases.
Suggestion: Please include a section in the paper that explicitly discusses the limitations of the SF-VAD method. For example, mention scenarios where the method might struggle (e.g., videos with very low-salience anomalies or those with high noise levels) and provide insights into why these limitations exist.
Impact on Evaluation: A thorough discussion of limitations will demonstrate a mature understanding of the method's scope and applicability. This could positively influence my evaluation score, especially if you also propose potential solutions or future work directions to address these limitations.
Code Organization and Documentation:
Description: The paper promises to release the data and code but does not mention the current state of the codebase or any efforts made to ensure its reproducibility.
Suggestion: Please organize and document the code following best practices for open-source projects. This includes providing a clear README file with installation instructions, usage examples, and a description of the code structure. Additionally, include comments within the code to explain key components and algorithms.
Impact on Evaluation: Well-organized and documented code will greatly facilitate the reproducibility of your work. If the code is released in a user-friendly manner, it could increase my evaluation score by demonstrating your commitment to open science and reproducibility.
Ablation Studies on Different Anomaly Types:
Description: The ablation studies section provides valuable insights into the effectiveness of different components of the proposed method. However, it does not specifically analyze the performance on different types of anomalies (e.g., subtle anomalies vs. obvious anomalies).
Suggestion: Please extend the ablation studies to include an analysis of the method's performance on different types of anomalies. This could involve categorizing anomalies based on their salience or other characteristics and evaluating how well the SF-VAD method performs on each category.
Impact on Evaluation: A detailed analysis of performance on different anomaly types will provide a more nuanced understanding of the method's strengths and weaknesses. This could positively influence my evaluation score if the method demonstrates robustness across a diverse range of anomalies.
Comparison with State-of-the-Art Methods on Additional Datasets:
Description: The paper compares the SF-VAD method with state-of-the-art methods on three datasets. However, it would be beneficial to see how the method performs on additional datasets to further validate its generalizability.
Suggestion: Please consider evaluating the SF-VAD method on additional publicly available video anomaly detection datasets. Compare its performance with existing methods on these datasets and discuss any differences in performance.
Impact on Evaluation: Demonstrating the method's generalizability across multiple datasets will strengthen its credibility and applicability. If the SF-VAD method consistently outperforms existing methods on additional datasets, it could significantly increase my evaluation score.

**Ethical Concerns:**

["NO or VERY MINOR ethics concerns only"]

**Final Justification:**

Thanks to the authors for their detailed responses. I basically acknowledge the contribution of this method. However, due to disagreements seen from comments and concerns of other reviewers, I would like to keep my rating score with lower confidence.

**Limitations:**

Potential Negative Societal Impacts:
Description: The paper does not explicitly discuss the potential negative societal impacts that may arise from deploying the SF-VAD method in real-world systems. This includes concerns related to privacy, surveillance, and potential biases in anomaly detection.
Suggestion: Add a dedicated section to discuss the ethical considerations and potential negative societal impacts of the SF-VAD method. Address issues such as privacy concerns (e.g., detecting behaviors that may be considered private or sensitive), surveillance implications (e.g., misuse in authoritarian regimes), and potential biases (e.g., racial, gender, or socioeconomic biases manifesting in anomaly detection). Provide guidelines or recommendations for mitigating these impacts.
Robustness Against Adversarial Attacks:
Description: The paper does not discuss the robustness of the SF-VAD method against adversarial attacks, which is an important factor to consider when deploying AI systems in security-critical applications.
Suggestion: Include an analysis of the method's robustness against adversarial attacks. This may involve testing the method on videos subjected to adversarial perturbations and discussing potential defense strategies or methods to enhance robustness.
Cross-Domain Generalization Capability:
Description: The paper primarily focuses on specific datasets and does not extensively discuss the generalization capability of the SF-VAD method across different domains or video types (e.g., surveillance footage, social media videos, medical videos).
Suggestion: Expand the evaluation to include diverse datasets from different domains. Discuss the challenges faced by the method in performing well across various video types and the potential adaptive adjustments that may be required.

**Paper Formatting Concerns:**

No formatting issues were detected; the figures and tables are clear, and the writing style is appropriate.

**Quality:**

3

**Strengths And Weaknesses:**

Strengths:
Potential to Address Practical Issues: The SF-VAD paradigm significantly enhances annotation efficiency while maintaining detection performance, which holds great significance in real-world applications where annotation resources are limited.
Rigorous Methodology: The paper proposes an innovative SF-VAD paradigm and effectively generalizes sparse frame-level supervision to event-level anomaly understanding through the FPL framework. The theoretical derivations and experimental validations are relatively rigorous, ensuring the reliability of the method.
Weaknesses:
Inadequate Discussion of Limitations: Although the paper highlights the advantages of the method, it fails to fully discuss its potential limitations or restrictions in applicable scenarios. For instance, the SF-VAD method may not perform well for certain specific types of anomalies.
Code Requires Organization and Standardization: While the paper promises to make the data and code publicly available, it is advisable to organize and standardize the code to ensure that other researchers can easily reproduce and utilize it. For example, following best practices of open-source projects, such as providing a clear README file, detailed comments, and documentation. Additionally, currently, there is no introduction to the dataset configuration in the paper, requiring reviewers to configure it themselves.

---

> ### Author Rebuttal · Authors · 2025-07-31
>
> We're sincerely grateful to the reviewer for their thorough assessment. Especially, we appreciate the affirmation of the **innovative SF-VAD paradigm** that addresses the practical challenges in VAD and the **effective FPL framework**.
> Here is our response to the reviewer's productive comment.
>
>
> ---
> **Response to Q1: Clarification on Dataset Configuration**
>
> We select the three most challenging VAD datasets, ShanghaiTech, UCF-Crime, and XD-Violence, that are well-recognized by the VAD community to validate the effectiveness of SF-VAD on a large-scale. The videos remain identical to those of the original datasets, with the single abnormal frame annotation being extended.
>
> **Data Splits**: we followed the original configurations for UCF-Crime and XD-Violence. For ShanghaiTech Campus, which was originally a semi-supervised dataset without abnormal videos in its training split, we adopted the widely applied data split created by Zhong et al. [1].
>
> **Dataset Annotation:** we adopt an authentic annotation protocol to thoroughly assess SF-VAD's effectiveness under practical conditions. Under this protocol, annotators were permitted to freely navigate within abnormal event occurrences to pinpoint the single abnormal frame. This approach maximized annotation efficiency in applied settings while inherently reflecting annotation bias. Further details can be found in Sec. B in Supp.
>
> **Data Preprocessing:** we adhere to a well-established paradigm. Initially, video frames are batched into 16-frame stacks. Subsequently, these frames are resized such that their shortest edge is 256 pixels, followed by 224-pixel crop augmentations. The processed video stacks are then input to a feature backbone for appearance or motion feature extraction. To manage sequence length variations, zero-padding or truncation is applied while ensuring the inclusion of the annotated abnormal frame.
>
> ---
> **Response to Q2: Discussion on Method Limitations**
>
> The discussion of limitations is provided in Sec. M of Supp. While our SF-VAD paradigm demonstrates robustness under varied scenarios, we acknowledge several promising avenues for future enhancement. The current multi-interval anomaly exploration primarily relies on individual video frame features; thus, an opportunity exists to further leverage inter-video correlations between abnormal annotations for richer pattern discovery. Additionally, integrating direct action traits could further enhance detection robustness, particularly for subtle anomalies. Due to the restrictions of the rebuttal policy, we are unable to provide specific failure case studies at this moment. However, we promise to include these in the revision.
>
> ---
> **Response to Q3: Code Organization and Documentation**
>
> Currently, a *README.md* file is included in the Supp. to provide an initial overview. Due to the rebuttal policy, we are unable to update code repository at this stage. However, upon publication, we are fully committed to releasing a user-friendly codebase, including the well-structured source code, detailed environment requirements and installation instructions, and all relevant annotation files that allow the VAD community to further investigate the potential of SF-VAD.
>
> ---
> **Response to Q4:  Generalization of Various Anomaly Types**
>
> To validate the generalizability of different anomaly types, we demonstrate performance across different types of anomalies in XD-Violence, as follows:
>
> |            | Arrest | Arson | Assault | Burglary | Vandalism | Explosion | Shooting |
> | ---------- | ------ | ----- | ------- | -------- | --------- | --------- | -------- |
> | UMIL       | 68.7   | 64.9  | 70.3    | 72.1     | 77.4      | 65.0      | -        |
> | RTFM       | 66.1   | 61.2  | 69.3    | 72.9     | 63.3      | 49.5      | 71.7     |
> | CoMo       | 56.2   | 47.5  | 74.1    | 69.4     | 83.8      | 64.6      | 75.1     |
> | FPL (Ours) | 70.8   | 71.7  | 81.5    | 72.3     | 85.8      | 70.3      | 82.9     |
>
> For further evaluation, we performed additional experiments for the class-wise AUC metric on UCF-Crime. The results are depicted as below:
> |           | Abuse | Explosion | Riot  | Fighting | overall |
> | --------- | ----- | --------- | ----- | -------- | ------- |
> | PEL4VAD   | 70.5  | 63.9      | 96.5  | 83.8     | 85.26   |
> | FPL(Ours) | 79.41 | 68.22     | 97.63 | 81.33    | 89.56   |
>
> Notably, our method surpasses state-of-the-art methods in a variety of abnormal categories. This includes subtle abnormal behaviors like _Arrest_ and _Assault_, which involve no dramatic scene changes, as well as more pronounced events such as _Explosion_ or _Arson_, where scenes typically involve dramatic changes and clear visual cues. Under both circumstances, our proposed SF-VAD paradigm and FPL method demonstrate compelling performance, which reaffirms the overall generalizability of the proposed methods. In contrast to previous methods [2, 3], which might adapt techniques like Gaussian splatting around the annotated frame, our approach ensures that FPL can adapt to various abnormal behaviors and dynamically explore the anomaly duration. We achieve this by adopting evidential learning to dynamically measure the relevance to the annotated abnormal behaviors and integrating frame-wise feature similarity accordingly to mine the most probable abnormal duration.
>
> ---
> **Response to Q5:  More Complete Evaluation**
>
> We appreciate the reviewer's valuable suggestion regarding additional evaluation. We contend that our current selection of three widely-used, real-world datasets, UCF-Crime, ShanghaiTech, and XD-Violence, was deliberate and has been widely used and thoroughly validated by previous VAD methods for comprehensive evaluation. These datasets are robust, encompassing diverse domains ranging from campus surveillance to real-world crime scenes and cinematic productions. They feature a **broad spectrum of anomaly behaviors**, include **scenarios with significant interference**, and cover **perspectives that vary from static to highly dynamic**. This comprehensive coverage enables a thorough assessment of our method's performance across various challenging conditions. More details can be found in Sec. E in Supp.
>
> **Authentic Annotation Protocol:** Our annotation process is designed to reflect genuine human interests in a realistic setting. The principle of our annotation protocol is to examine the effectiveness of SF-VAD in practical scenarios, where annotators are allowed to freely navigate the video timeline and locate a random abnormal frame as soon as they witness an anomaly, without any preference guidance. Such a principle ensures that the annotation evades prior guidance that could artificially inflate performance. The detailed description of the annotation process and requirements has been provided in Sec. 4.1 and Sec. B in Supp., alongside dataset statistics in Sec. 4.2 and Sec. C in Supp.
>
> **Cross-Dataset Transferability:** To further validate SF-VAD's generalizability and practical feasibility, we specifically examined its **cross-dataset transferability**.
> | Source     | UCF-Crime | XD-Violence       | UCF-Crime   | XD-Violence       | ShanghaiTech | UCF-Crime        | XD-Violence |
> | ---------- | --------- | ----------------- | ----------- | ----------------- | ------------ | ---------------- | ----------- |
> | Target     | →         | UCF-Crime         | XD-Violence | ←                 | →            | ShanghaiTech     | ←           |
> | RTFM       | 84.48     | 68.59 (18.81% ↓)  | 76.62       | 37.30 (51.32% ↓)  | 97.20        | 45.49 ( 53.2% ↓) | -           |
> | CoMo       | 86.07     | 69.89 ( 18.80% ↓) | 81.31       | 46.74 ( 42.52% ↓) | 97.59        | 52.02 ( 46.7% ↓) | -           |
> | FPL (Ours) | 90.23     | 85.67 (5.05% ↓)   | 89.56       | 82.31 (8.10% ↓)   | 98.32        | 94.21 (4.18% ↓)  | -           |
>
> The results indicate a commendable cross-dataset transferability. This is primarily because single-frame supervision provides an unambiguous definition of the anomaly, ensuring a clean and precise learning target that is less susceptible to noise and irrelevant contextual information, which aids in generalizing across varied datasets.
>
> **Computation Efficiency:** Furthermore, our evaluation included assessing the method's computational efficiency.
> || Supervision| Parameters (↓) | Inference Time (↓) | UCF-Crime AUC |
> |-|-|-|-|-|
> |Conv-AE|Semi-Supervised|10.14 M|-|-|
> |MGFN| Weakly-Supervised|28.46 M|3.70 s|86.67|
> |RTFM| Weakly-Supervised|24.72M|3.23 s|84.30|
> |UR-DMU|Weakly-Supervised| 6.49 M|12.39 s|86.97|
> |VadCLIP|Weakly-Supervised (with Text Features) |162.26 M|5.65 s|88.02
> | **FPL(Ours)** | Single-Frame Supervised|**2.21 M**|**0.51 s**|**90.23**|
>
> Remarkably, our SF-VAD method demonstrates a **smaller parameter size**, much **faster inference time**, and **better detection performance** than previous state-of-the-art semi-supervised and weakly-supervised VAD methods. This superior performance-cost trade-off stems from the precise abnormal learning target enabled by FPL, which integrates anomaly relevance estimation with feature similarity to deduce more accurate learning targets than the top-k selection in MIL, which is less robust to noise interference. As a result, it allows a lightweight network architecture, without computationally expensive memory modules or text encoders, to effectively capture diverse anomalous cues.
>
> ---
> [1] Zhong, Jia-Xing, et al. "Graph convolutional label noise cleaner: Train a plug-and-play action classifier for anomaly detection." *CVPR*, 2019.
>
> [2] Zhang, Huaxin, et al. "Glancevad: Exploring glance supervision for label-efficient video anomaly detection." *arXiv*, 2024.
>
> [3] Cui, Ran, et al. "Video moment retrieval from text queries via single frame annotation." *SIGIR*, 2022.

---

> ### Comment · Area_Chair_UFE4 · 2025-08-07
> **Discussion**
>
> Dear Reviewers,
> Could you please review the authors’ rebuttal and let them know if you have any further comments or concerns? Do you feel your original comments have been adequately addressed? If not, it would be helpful to highlight any remaining issues.
> I would be grateful if you could kindly engage in the discussion to help move the review process forward.
> Best regards,
> AC

---

> > ### Comment · Area_Chair_UFE4 · 2025-08-08
> > **Discussion**
> >
> > Dear Reviewers, Could you please review the authors’ rebuttal and let them know if you have any further comments or concerns? Do you feel your original comments have been adequately addressed? If not, it would be helpful to highlight any remaining issues. I would be grateful if you could kindly engage in the discussion to help move the review process forward. Best regards,
> > AC

---

### Note · Authors · 2025-08-14

We express our sincere gratitude to the Area Chair for their diligent efforts in facilitating a productive discussion, as well as to all the Reviewers for their insightful and constructive reviews. We are delighted to hear that our paper introduces an **innovative and rational SF-VAD paradigm** that effectively addresses the practical challenges in VAD (Reviewers BiMk, ZVwY, and mbCU), and that the **Frame-guided Progressive Learning (FPL) framework is effective and robust** in practical settings (Reviewers BiMk and mbCU). In response to the Reviewers' valuable feedback, we have thoroughly validated and clarified our method from the following perspectives:

- We added further evaluation of the **generalizability**, including performance comparison across various domains, diverse anomaly types, multi-event scenarios, shifting annotations, lightweight backbones, and cross-dataset transferability (Reviewer 2TVV’s Q4; Reviewer mbCU’s Q1, Q5, Q6). SF-VAD demonstrates compelling robustness under various conditions, providing a feasible solution for real-world applications.
- We substantiated the strong **efficiency** of SF-VAD, both in terms of annotation cost (Reviewer BiMk’s Q1) and computational overhead (Reviewer BiMk’s Q5; Reviewer ZVwY’s Q5; Reviewer 2TVV’s Q6).
- We elaborated the **advantages** of our approach over existing paradigms, demonstrating how it solves the problem of lacking an abnormal behavior reference in coarse-grained supervision (Reviewer ZVwY‘s Q2; Reviewer 2TVV’s further response), mitigates annotation bias, and reduces the burden of exhaustive boundary labeling. As a result, SF-VAD delivers a favorable performance-cost trade-off.
- We further clarified the **motivation** and operational mechanisms of FPL (Reviewer ZVwY’s Q1, further response; Reviewer mbCU’s Q4), providing a more solid theoretical foundation for its introduction and design.

In conclusion, SF-VAD leverages **efficient** frame annotation by exploring anomaly relevance and feature similarity, utilizing a **lightweight** network to demonstrate **superior performance and generalizability** across diverse scenarios, offering a novel and effective paradigm for VAD.
We kindly request that Area Chair and Reviewers take into account the contributions and clarifications of our work in making the final decision.

---

### Decision · Program_Chairs · 2025-09-17

**Decision:**

Accept (poster)

**Comment:**

The reviewers and AC agree that the paper demonstrates sufficient novelty, presents promising results, and should be accepted. The Area Chair recommends accepting the paper and encourages the authors to incorporate the reviewers' comments in the final version.